# From paleness to albinism: Contribution of *OCA2* exon 10 skipping to hypopigmentation

Elina Mercier[1], David-Alexandre Trégouët[2], Sébastien Campagne[3], Vincent Michaud[1,4], David J. Green[5], Shahram Mesdaghi[6,7], Orane Le Gallais[1], Orphée Jouannigot-Castano[1], Daniel J. Rigden[6], Catherine Estay-Ahumada[8], David Hicks[8], Panagiotis I. Sergouniotis[5], Benoit Arveiler[1,4], Sophie Javerzat[1]*

1 Rare Diseases Genetics and Metabolism, INSERM U1211, University of Bordeaux, Bordeaux, France, 2 Bordeaux Population Health Research Center, UMR 1219, University of Bordeaux, Inserm, Bordeaux, France, 3 CNRS, INSERM, ARNA, UMR 5320, U1212, University of Bordeaux, Bordeaux, France, 4 Molecular Genetics Laboratory, Bordeaux University Hospital, Bordeaux, France, 5 Division of Evolution, Infection and Genomics, School of Biological Sciences, Faculty of Biology, Medicine and Health, University of Manchester, Manchester, United Kingdom, 6 Department of Biochemistry, Cell and Systems Biology, Institute of Systems, Molecular & Integrative Biology, Liverpool, United Kingdom, 7 Computational Biology Facility, MerseyBio, University of Liverpool, Liverpool, United Kingdom, 8 Institute for Cellular and Integrative Neuroscience, CNRS UPR 3212, Strasbourg, France

* sophie.javerzat@u-bordeaux.fr

## Abstract

Pathogenic variants in the *OCA2* gene result in oculocutaneous albinism. In humans and several other mammalian species, *OCA2* is transcribed into two coding mRNAs, a major transcript that encodes the full-length protein and a minor transcript that skips in-frame exon 10 and has no identified function. By contrast, rodents and non-mammalian vertebrates only express the major transcript. Several rare variants of human *OCA2* have been reported to be pathogenic due to increased exon 10 skipping. To understand the vulnerability of human *OCA2* to exon 10 skipping and its impact on pigmentation, we first functionally tested a series of rare missense variants spread across exon 10. We found that each variant significantly influences the skipping ratio either positively or negatively. By combining human and murine sequences in functional assays, we found that the skipping ratio depends on particular exonic and intronic combinations. Next, we focused on the most frequent exonic single nucleotide variant of human *OCA2*, rs1800404-T (c.1065G>A/p.Ala355=), located in exon 10. We show that this variant significantly promotes exon 10 skipping on its own and exhibits an additive effect in *cis* to previously identified splicing variants likely contributing to their pathogenicity. Association studies reveal that rs1800404-T is tightly associated with lighter skin and hair pigmentation in a representative European population, as has been observed in other populations around the world. Our structural models of the skipped transcript-derived protein support causality by predicting that it could exert a dominant-negative effect, which is consistent with a dose-dependent hypopigmentation response in both pathological and physiological

**Data availability statement:** The new variant identified in this study has been uploaded to ClinVar public database (SCV006311562). Data for building Figs 1, 3, 4, 5 and related statistical analysis can be accessed at https://doi.org/10.5281/zenodo.15829093 AlphaPulldown output files for the OCA2 and ΔE10-OCA2 can be accessed at https://doi.org/10.5281/zenodo.15828886 UK Biobank data are available under restricted access through a procedure described at http://www.ukbiobank.ac.uk/using-the-resource/. This research was conducted using the UK Biobank Resource under project 53114. All other data supporting the findings of this study are available within the article (including its Supporting information files).

**Funding:** Genespoir, the French albinism association to SJ; the French National Research Agency / Agence Nationale de la Recherche (ANR-21-CE17-0041-01 to BA); the Wellcome Trust (224643/Z/21/Z to P.I.S.); the UK National Institute for Health Research (NIHR) Clinical Lecturer Programme (CL-2017-06-001 to P.I.S.); the NIHR Manchester Biomedical Research Centre (NIHR 203308 to P.I.S.). The funders had no role in study design, data collection and analysis, decision to publish, or preparation of the manuscript.

**Competing interests:** The authors have declared that no competing interests exist.

contexts. Overall, the modulation of *OCA2* exon 10 skipping by both benign and pathogenic variants provides a basis for improving the genetic diagnosis of albinism, and paves the way for research into the molecular and evolutionary mechanisms behind human pigmentation diversity.

## Author summary

The loss of function of *OCA2*, a key player in pigmentation, results in oculocutaneous albinism. In humans and several other mammalian species, an alternative transcript lacking exon 10 is expressed at low levels. We have discovered that the levels of exon 10 skipping are tightly modulated by both benign and pathogenic variants. This results in a dose-dependent association with pigmentation across a continuous spectrum, ranging from physiological skin and hair color variation to albinism. Our study identifies the most frequent *OCA2* single nucleotide variant (SNV), rs1800404 (c.1065G/A), as a causal variant that is likely to impact melanogenesis levels in both pathological and physiological contexts, depending on the haplotype into which it is embedded. This article provides significant and novel insight into the modulation of exon skipping, which is of broad interest for improved genetic diagnosis as well as for the understanding of the complex genetic determinism of pigmentation. Furthermore, this study opens avenues for elucidating the variability of phenotypes that depend upon pigmentation, such as retinal development and visual acuity.

## Introduction

Like most genetic conditions, albinism displays a wide range of phenotypic manifestations, which can make diagnosis challenging. Clinical diagnosis predominantly relies on quantitative assessment of specific ophthalmological abnormalities as well as skin and hair hypopigmentation - although the latter may go unnoticed [1,2]. Pathogenic variants that cause albinism express themselves as a continuous spectrum from complete lack of pigmentation and severe visual impairment to marginally subnormal vision and mild sensitivity of the skin to UV-light. Notably, most genes involved in non-syndromic oculocutaneous albinism (OCA) are also associated with fine-tuned modulation of pigmentation in healthy individuals. This results in single nucleotide variants (SNVs) in these genes being exploited in forensic DNA profiling to predict skin, hair and eye colour [3,4]. Of note, a quarter of the SNVs selected in the last version of the multiplex forensic phenotyping toolkit HirisPlex-S (10 out of 41), are in *OCA2* or in its enhancer sequence located in *HERC2* [5], illustrating a major role of *OCA2* in modulating pigmentation.

  *OCA2* encodes the OCA2 protein (previously called P protein), a melanosomal transmembrane anion channel that is instrumental in regulating pH within the pigment synthesizing organelle [6]. Its contribution to melanogenesis appears equivalent in

the retinal pigment epithelium (RPE) and in cutaneous melanocytes [7]. Lack of OCA2 protein markedly impairs melanogenesis mediated by the three tyrosinase-related enzymes, Tyrosinase, Tyrosinase-related protein-1 and Dopachrome tautomerase (Tyrosinase-related protein-2), respectively encoded by *TYR*, *TYRP1* and *DCT/TYRP2*. Of note, the OCA2 protein has been reported to have a role in the processing and trafficking of Tyrosinase from the endoplasmic reticulum to the melanosome [8]. Loss-of-function mutations in *OCA2* thus have a direct epistatic effect on the Tyrosinase encoding gene *TYR*. Once in the melanosome, provided the pH is optimally controlled by several channels including the OCA2 protein, Tyrosinase catalyzes the initial and rate-limiting steps of the melanogenic biosynthetic pathway by converting tyrosine into dopaquinone [9]. This releases L-Dopa, a major factor implicated in retinal development. L-Dopa is abundantly produced in the embryonic RPE at the onset of melanogenesis and triggers critical steps in early retinal development [10,11]. Hence, individuals with *TYR*- and *OCA2*-related albinism exhibit comparable levels of visual impairment and overlapping skin and hair pigmentation phenotypes [1].

Pathogenic variants in *OCA2* account for ~30% of genetic diagnosis of patients with clinically confirmed albinism [12]. The major transcript of *OCA2* carries all 24 exons. An alternative transcript that lacks the in-frame exon 10 is expressed at low levels in healthy individuals, although to date, the corresponding putative protein remains undescribed. In a previous study we showed that several rare variants including synonymous variants of exon 10 significantly increase exon 10 skipping resulting in albinism [13]. Additional missense variants have been reported in exon 10 and are classified as variants of unknown significance (VUS) due to poor *in silico* prediction of protein misfolding. Their potential effect on exon 10 skipping has not been investigated although splicing alterations are frequently triggered by variants that lie in so-called vulnerable exons [14]. These effects are however difficult to predict *in silico*, reflecting the complexity of splicing *cis*-regulation [15]. OCA genes do not escape the rule, with an increasing number of splice variants being identified by functional assays that help resolve the genetic diagnosis of patients [16,17].

Interestingly, basal levels of *OCA2* exon 10 skipped transcripts are only seen in some mammalian species such as primates, but not in rodents. This contrasts with the structure/function similarity of the orthologous OCA2 protein in human and mouse, that has made mouse mutant lines and cell-derived models overexpressing the human OCA2 protein so useful to dissect its roles in melanogenesis [6,18]. It also questions a possible role of the skipped transcript in primates and other mammals that express basal levels of this alternative mRNA.

Here, we investigated rare missense VUS that are distributed all along *OCA2* exon 10 and evaluated their effect on splicing. We then asked what makes human *OCA2* highly vulnerable to exon 10 skipping. Using different combinations of mouse and human constructs, we investigated the sequences that are involved in the control of exon retention/skipping. We next focused on the most frequent exonic variant of *OCA2* c.1065G>A/p.Ala355= (rs1800404-T) that lies in exon 10 and has previously been described as associated with increased levels of skipped transcripts [19] and with lighter pigmentation in Africans and other populations with dark skin [19–23]. Here, we show that this variant enhances exon 10 skipping in a minigene assay and demonstrate its robust association with skin and hair paleness in the European population. These data are of interest not only in the context of genetic characterization of albinism and diagnosis, but also as a starting point for understanding the molecular and evolutionary mechanisms behind the physiological diversity of pigmentation in humans.

## Results

### Rare missense variants of *OCA2* exon 10 affect skipping profile

Six missense VUS of exon 10 (Table 1) that had been reported in different studies or identified in our OCA patient database were tested by minigene assay. Variant c.1048G>A/p.Val350Met has been reported twice in individuals with albinism [24,25] and, at the time of writing, is classified VUS in ClinVar. In our local database, this variant was found once in a Cypriot patient presenting with albinism who also carried a deep intronic *OCA2* VUS (c.2080-11930G>T, absent from

**Table 1. In silico predictions and allelic frequencies of missense variants included in this study.**

| OCA2 variants | | In silico predictions | | | | | GnomAD allele frequency (number HTZ and HMZ) | | |
|---|---|---|---|---|---|---|---|---|---|
| c. | p. | SPiP | SpliceAI | Pangolin | Polyphen 2 | Alphamissense | v3.1.2 | v4.1.0 (non-UKB) | ClinVar report and variation ID |
| 1048G>A | Val350Met | 09.63% (no effect) | AL=0.02, 3 bp (below threshold) | 0.05 (below threshold) | 0.984 (supporting pathogenic) | 0.3939 (indeterminate) | 9.2E-05 (14 HTZ-0 HMZ) | 1.5E-04 (115 HTZ-2 HMZ) | VUS 1424156 |
| 1061T>C | Leu354Pro | 09.76% (no effect) | AG=0.01, 16 bp (below threshold) | 0.05 (below threshold) | 0.997 (supporting pathogenic) | 0.8837 (supporting pathogenic) | 6.6E-06 (1 HTZ-0 HMZ) | 2.6E-06 (2 HTZ-0 HMZ) | VUS 2481092 |
| 1064C>T | Ala355Val | 30.67% (ESR element alteration) | AL=0.01, 19 bp (below threshold) | 0.04 (below threshold) | 0.981 (supporting pathogenic) | 0.602 (indeterminate) | 5.9E-05 (9 HTZ-0 HMZ) | 2.3E-04 (181 HTZ-4 HMZ) | VUS 617804 |
| 1081C>G | Leu361Val | 03.59% (no effect) | AG=0.01, 36 bp (below threshold) | 0.05 (below threshold) | 0.9 (indeterminate) | 0.178 (indeterminate) | 6.6E-06 (1 HTZ-0 HMZ) | 1.3E-05 (10 HTZ-0 HMZ) | Absent |
| 1109T>C | Ile370Thr | 05.05% (no effect) | DL=0.00, -7 bp (no effect) | 0.02 (below threshold) | 0.038 (supporting benign) | 0.0781 (moderate benign) | 3.3E-03 (508 HTZ-4 HMZ) | 9.7E-04 (755 HTZ-4 HMZ) | VUS Benign Likely benign 419971 |
| 1114G>A | Asp372Asn | 09.3% (no effect) | DG=0.01, -2 bp (no effect) | 0.01 (below threshold) | 0.086 (supporting benign) | 0.0996 (moderate benign) | 2.6E-05 (4HTZ-0 HMZ) | 2.0E-05 (16 HTZ-1 HMZ) | VUS 2168452 |

Nomenclature based on NM_000275.3. Predictions and frequencies gathered from mobiDetails SPiP: https://mobidetails.iurc.montp.inserm.fr/MD/; SpliceAI (unmasked, max distance 500 bp), Pangolin: https://spliceailookup.broadinstitute.org; Alphamissense: https://alphamissense.hegelab.org/; gnomAD: https://gnomad.broadinstitute.org/; ClinVar: https://www.ncbi.nlm.nih.gov/clinvar/.

Abbreviations: HMZ: homozygous; HTZ: heterozygous; UKB: UK Biobank; AG:acceptor gain; AL: acceptor loss; DG: donor gain; DL: donor loss.

ClinVar) in *trans* and no pathogenic variants in any albinism genes. Variant c.1061T>C/p.Leu354Pro appeared only once in ClinVar as VUS. Variant c.1064C>T/p.Ala355Val has been reported once in a patient with albinism [26]. It has been classified VUS twice and likely pathogenic once in ClinVar. Variant c.1081C>G/p.Leu361Val has been identified in one patient of our cohort who carried a deep intronic *OCA2* VUS (ClinVar ID: 3256646) in *trans*. Variant c.1109T>C/p.Ile-370Thr has been reported in two patients with albinism [24] and recorded as of Uncertain Significance, Benign or Likely Benign in ClinVar. Variant c.1114G>A/p.Asp372Asn is reported once in ClinVar as a VUS.

The positions of these variants are indicated in Fig 1a together with those of the nine variants functionally tested in our previous study [13]. Allelic frequencies, *in silico* predictions of splicing dysregulation or protein conformational changes as well as ClinVar reports are provided in Table 1.

Worldwide allele frequencies were notably high for some variants, with several homozygotes being reported for c.1048G>A, c.1064C>T, c.1109T>C. Overall, only c.1064C>T/p.Ala355Val displayed a significant score of potential alteration of a splicing regulatory element. As for protein prediction, *in silico* analysis supported likely pathogenicity for c.1061T>C/p.Leu354Pro only.

Results of the minigene assay are presented in Fig 1b, 1c. The common benign variant c.1065G>A/p.Ala355= (highlighted in Fig 1a) was included in all constructs because it was in *cis* to the tested missense variant in the corresponding patients from our cohort. In order to optimize quantification of the skipping ratio, these assays were repeated 4 times in independent experiments and the RT-PCR products run on an automated electrophoresis device coupled to data analysis software. Details of the steps in data processing are provided in S1 Fig. As illustrated, each of the six missense variants

showed a significant effect on splicing compared to the wild-type sequence taken as a control (17% of skipping). Only variant c.1061T>C/p.Leu354Pro resulted in better retention of exon 10 compared to control. All other missense variants significantly increased the percentage of skipping but to different extents. For further interpretation, these levels were compared to the profile of synonymous variant c.1080C>T/p.Ser360= (65% skipping), previously shown as pathogenic due to excessive skipping [13]. Among the newly tested missense variants, only c.1081C>G/p.Leu361Val exceeded this threshold with 87% of skipped products. Variant c.1048G>A/p.Val350Met reached 53% of skipping which is likely to contribute to its pathogenicity. The three remaining missense variants showed weaker levels of enhancement reaching around 28% of skipping each. Of note, despite lying 3 nucleotides before the exon/intron junction, c.1114G>A/p.Asp372Asn had the weakest impact on loss of retention. Overall, the optimized minigene assay strongly supports pathogenicity primarily due to splicing defect for c.1081C>G/p.Leu361Val. On the other hand, variant c.1061T>C/p.Leu354Pro shows less skipping than the reference sequence, and its pathogenicity is likely to be the consequence of deleterious protein conformation changes as suggested by the *in silico* analysis (Table 1).

### *OCA2* exon 10 skipping is the hallmark of primates and some mammalian species but not rodents

All six single nucleotide substitutions tested above together with the 2 synonymous variants previously described are evenly distributed along the 72 bp of exon 10 and have a significant effect on skipping despite being more than 2 bp away from the intron/exon junction (Fig 1a).

The alternative transcript resulting from *OCA2* exon 10 skipping is reported only in some mammalian species. We searched the sequence databases for *OCA2* exon 10 skipped transcripts in a series of mammals and could identify the alternative skipped transcript in primates and representatives of different mammalian groups (carnivores, laurasiatherian, artiodactyla, and glires) including the rabbit but not rodents such as mouse or rat (S2 Table). Alignments of the exon 10 sequences from selected representatives are provided in S2 Fig. Alignment of the human and murine exon 10 sequences highlights 12 nucleotide differences in the 72 bp exonic sequence (Fig 2a). Most of these differences are synonymous. Only two codons are different, at position p.360, with Ser(TCC) in human and Ala(GCA) in the mouse, and at position p.370, with Ile(ATT) in human and Val(GTT) in the mouse. Of note, (A) lies at position c.1065 which corresponds to the common European human variant c.1065G>A.

The splicing profile of exon 10 was further analyzed by RT-PCR on C57BL/6J mouse skin RNA and compared to the human splicing profile. Only one form of RT-PCR product including exon 10 was obtained from mouse tissue, confirming the total retention of exon 10 in this rodent species (Fig 2b). The absence of detectable exon 10 skipping was also verified by testing RNA extracted from mouse skin at different ages as well as pigmented optic cup samples (S3a Fig).

These findings prompted us to test the murine sequence in the minigene assay. As shown in Fig 2c, RT-PCR performed on RNA extracted from cells transfected with the murine sequence homologous to the human construct (exon 10 surrounded by 335 bp of intron 9 and 313 bp of intron 10), only amplifies a single band corresponding to the correctly spliced RNA with exon 10. The mouse minigene construct therefore contains the minimal sequences for total exon retention and recapitulates the murine *in vivo* profile.

We then used the human and murine minigene constructs to test reciprocal exonic mutations as summarized in Fig 2d (see S3b, S3c Fig for experimental data). Three codons of exon 10 were specifically targeted. The first codon p.349 encodes Ile in both human and mouse but with a different third nucleotide, ATC/T (C in human, T in mouse). The human synonymous variant c.1047C>T/p.Ile349= (Fig 1a) therefore corresponds to the murine codon (ATT). However, within the human context it significantly increased skipping [13], whereas no skipping was detected with the reciprocal change, i.e., (ATC) in the murine context. This indicates that the effect of this nucleotide change is dependent upon the surrounding sequences. Codon sequence at position p.360 was reciprocally swapped. At this position the murine codon Ala(GCA) replacing Ser(TCC) in the human minigene resulted in a marked increase in skipping whereas the reciprocal sequence did not induce skipping of the murine exon 10. None of the nucleotide changes that were introduced in this codon resulted in

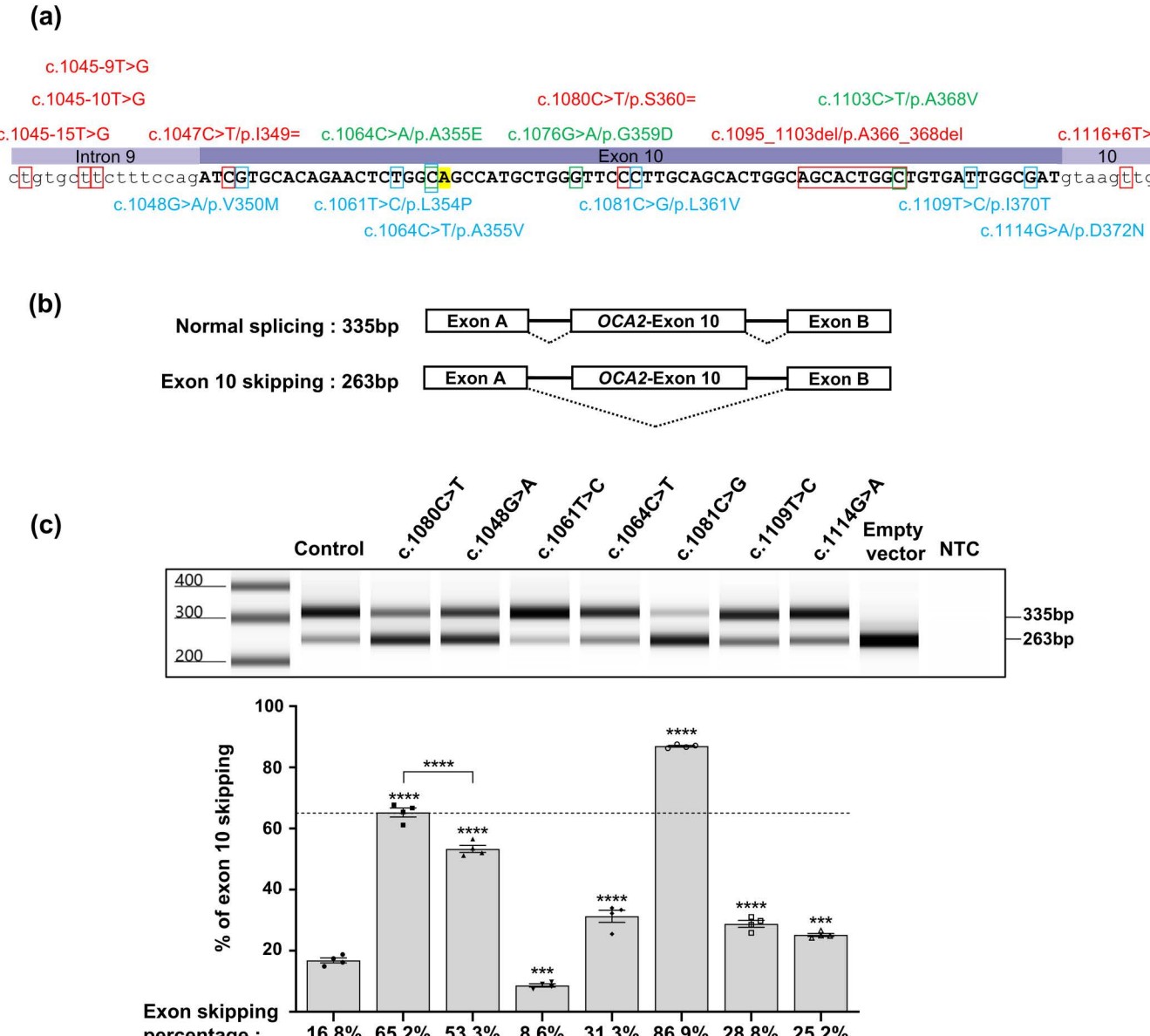

**Fig 1. Minigene assay of rare missense VUS in exon 10. (a)** Position of variants that were tested by minigene assay. Variants in red have been shown pathogenic due to excessive exon skipping in our previously reported study [13] whereas variants in green had no significant effect. Missense variants tested in (c) are indicated in blue. The position of the frequent benign variant c.1065G>A/p.A355= (rs1800404:C/T) is highlighted in yellow. Note that all rare variants were tested in *cis* to c.1065G>A which is predominant in European populations and in our patient cohort. **(b)** Minigene construct and expected sizes of the RT-PCR products with (335 bp) or without (263 bp) exon 10. **(c)** Representative electrophoresis profile of RT-PCR products from variants tested in minigene assay compared to control and pathogenic synonymous variant c.1080C>T; p.Ser360=, using Agilent 4150 TapeStation instrument and D1000 DNA ScreenTape system. NTC: no-template control. Relative quantification of exon 10 skipping in 4 series of independent experiments is represented by histograms below the image. Mean % of skipping is indicated below each histogram. All of the tested variants significantly increase exon 10 skipping except for c.1061T>C; p.Leu354Pro which decreased skipping. Error bars represent SEM (n = 4). *** indicates p < 0.001, **** indicates p < 0.0001, one-way ANOVA followed by the Bonferroni post hoc multiple comparison test.

**(a)**

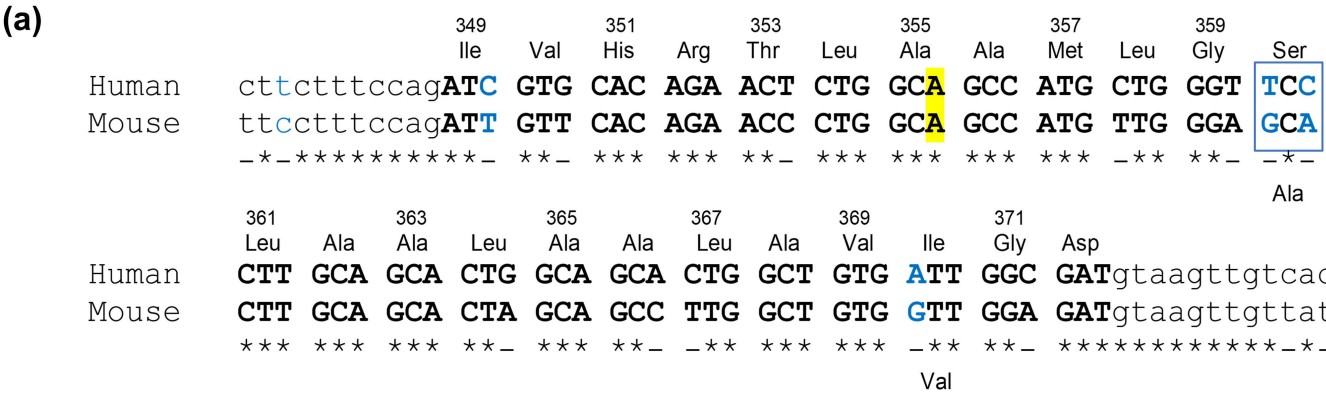

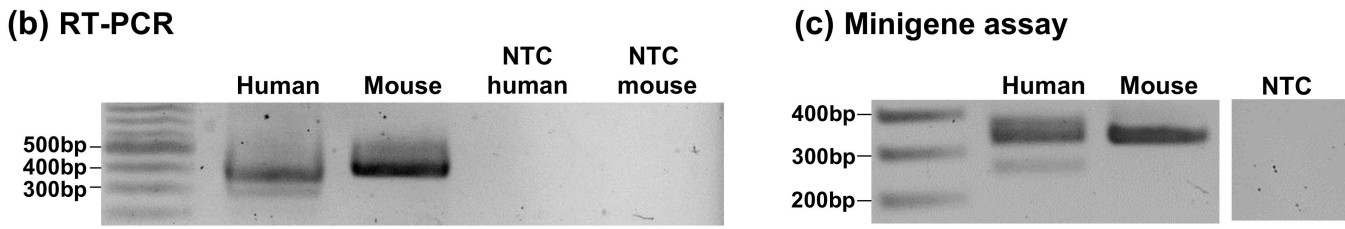

**(b) RT-PCR**

**(c) Minigene assay**

**(d)**

| | Human | Skip | Mouse | Skip |
|---|---|---|---|---|
| p.349 WT | Ile (ATC) | 15-20% | Ile (ATT) | No skip |
| | Ile (ATT) | **>40%** | Ile (ATC) | No skip |
| p.360 WT | Ser (TCC) | 15-20% | Ala (GCA) | No skip |
| | Ser (TCA) | **>30%** | Ser (TCA) | No skip |
| | Ala (GCC) | **>70%** | Ser (TCC) | No skip |
| | Ala (GCA) | **>80%** | | |
| p.370 WT | Ile (ATT) | 15-20% | Val (GTT) | No skip |
| | Val (GTT) | 15-20% | Ile (ATT) | No skip |
| p.360+p.370 WT | TCC+ATT | 15-20% | GCA+GTT | No skip |
| | | | TCC+ATT | No skip |

**Fig 2. Minigene assay profiling of human and mouse combined exon 10 sequences. (a)** Alignment of human and mouse *OCA2* exon 10 (upper-case) and flanking intronic (lowercase) sequences (human: ENSG00000104044; mouse: ENSMUSG00000030450) showing high exon sequence homology (~82%). Sequences alignment was performed on Clustal Omega (https://www.ebi.ac.uk/jdispatcher/msa/clustalo). Conserved nucleotides are indicated with an asterisk. Amino acids and their position in the human protein are indicated above the alignment. The two amino acid that are different

in mouse are indicated below. Nucleotides indicated in blue have been targeted for minigene constructs. The position of the frequent benign variant c.1065G>A/p.A355= (rs1800404:C/T) is highlighted in yellow. **(b)** RT-PCR products using RNAs from human melanocytes or mouse skin and primers in *OCA2* exon 9 and 13 (see Methods). The two bands from the human sample correspond to the two transcripts with (383 bp) or without (311 bp) exon 10, whereas in mouse all transcripts retain exon 10 resulting into a single RT-PCR band (424 bp). NTC = no-template control. **(c)** Human and mouse splicing profile by minigene assay. The two bands from the human minigene correspond to the 2 transcripts with (335 bp) or without (263 bp) exon 10, whereas with the mouse gene derived construct, all transcripts retain exon 10 resulting into a single RT-PCR band (335 bp). **(d)** Summary of the effect of nucleotide swapping between human and mouse exon 10 sequences in the minigene assay. Minigene constructs with a human backbone are framed in purple. Minigenes with a mouse backbone are framed in orange. The changes are highlighted using the corresponding colours, experimental data in S3 Fig.

preventing skipping of the human minigene or induced skipping of the murine minigene. Swapping of the codon sequence at p.370 with Val(GTT) in the human minigene or Ile(ATT) in the mouse minigene had no effect, nor did the introduction of both human p.360 and p.370 sequences in the murine minigene.

## Vulnerability of human *OCA2* to exon 10 skipping relies on the combination of exonic and intronic sequences

As none of the single nucleotide changes that we tested in the human exon sequence prevented its skipping, we next questioned the role of the surrounding *OCA2* introns by assaying chimeric constructs made of human and murine sequences (Fig 3). As illustrated in Fig 3a, when human exon 10 was flanked by mouse intron 9 and 10 sequences (~300 bp each), full retention was observed as with the murine control minigene. The reciprocal construct (murine exon 10 flanked by human intronic sequences) also resulted in full retention. The presence of only one of the segments (intron 9 or exon 10 or intron 10) of murine origin was sufficient to protect exon 10 from skipping. Moreover, in the murine minigene, the intronic segments could be shortened down to 75 bp for intron 9 (including the putative branch point at -37 and the polypyrimidine track) and 20 bp for intron 10 without affecting exon 10 retention (Fig 3b). These results indicated that human exon 10 vulnerability is dependent upon a combination of intronic and exonic sequences.

We next investigated in more detail the potential role of nucleotide at position c.1045-9 which is the first divergent nucleotide between the human and murine sequences upstream the intron 9/exon 10 junction (Fig 2a). At this position, (T) is associated with basal levels of skipping (human reference sequence), whereas (G) is pathogenic as it causes full skipping [13]. The mouse sequence carries a (C) and is associated to full retention in the murine context. Nucleotide c.1045-9G was introduced in the murine and chimeric constructs, and as shown in Fig 3c, this variant induced skipping only when embedded in an intron 9 sequence of human origin.

Sequence homology in this region was next analyzed in a series of mammalian species which showed identity of the intron 9 sequence from c.1045-18 to the intron/exon junction among non-human primates that express the skipped transcript (S2 Table, S2a Fig). Among non-primate mammalian species, only the rodent *Psammomys obesus* was found to have a 17 bp identical sequence in this region (S4a Fig). *Psammomys obesus* (also known as Fat Sand Rat) is a diurnal gerbil with a pale coat colour that has proved a valuable model for studying several eye diseases such as diabetic retinopathy and Stargardts disease [27,28]. The unexpected *OCA2* intron 9 sequence homology between this rodent and primates prompted us to look at the splicing of its exon 10. RNA was purified from skin and pigmented optic cups of adults and used for RT-PCRs (S4b Fig). The data show that exon 10 is fully retained in *Oca2* mRNA of *Psammomys* despite a much higher homology to human than mouse or rat in the flanking intron 9 sequence.

The sequence corresponding to the shortened mouse minigene was next examined in selected mammalian species that do (human and non-human primates, lion, blue whale, donkey, rabbit) or do not allow exon 10 skipping (mouse, rat, *Psammomys*) (S2 Table; S2 Fig). This revealed a stretch of ~40 bp, -20 to -60 bp upstream of the intron 9 3′ splice site of species vulnerable to exon 10 skipping (including the rabbit), which is absent in mouse, rat and *Psammomys*. In addition, alignment of the 20 bp of intron 10 included in the shortened mouse minigene showed that the sequence (+ 1 to + 8) recognized by the U1snRNA [29] is identical in all studied species whether exon 10 is vulnerable to skipping or not. From

PLOS Genetics

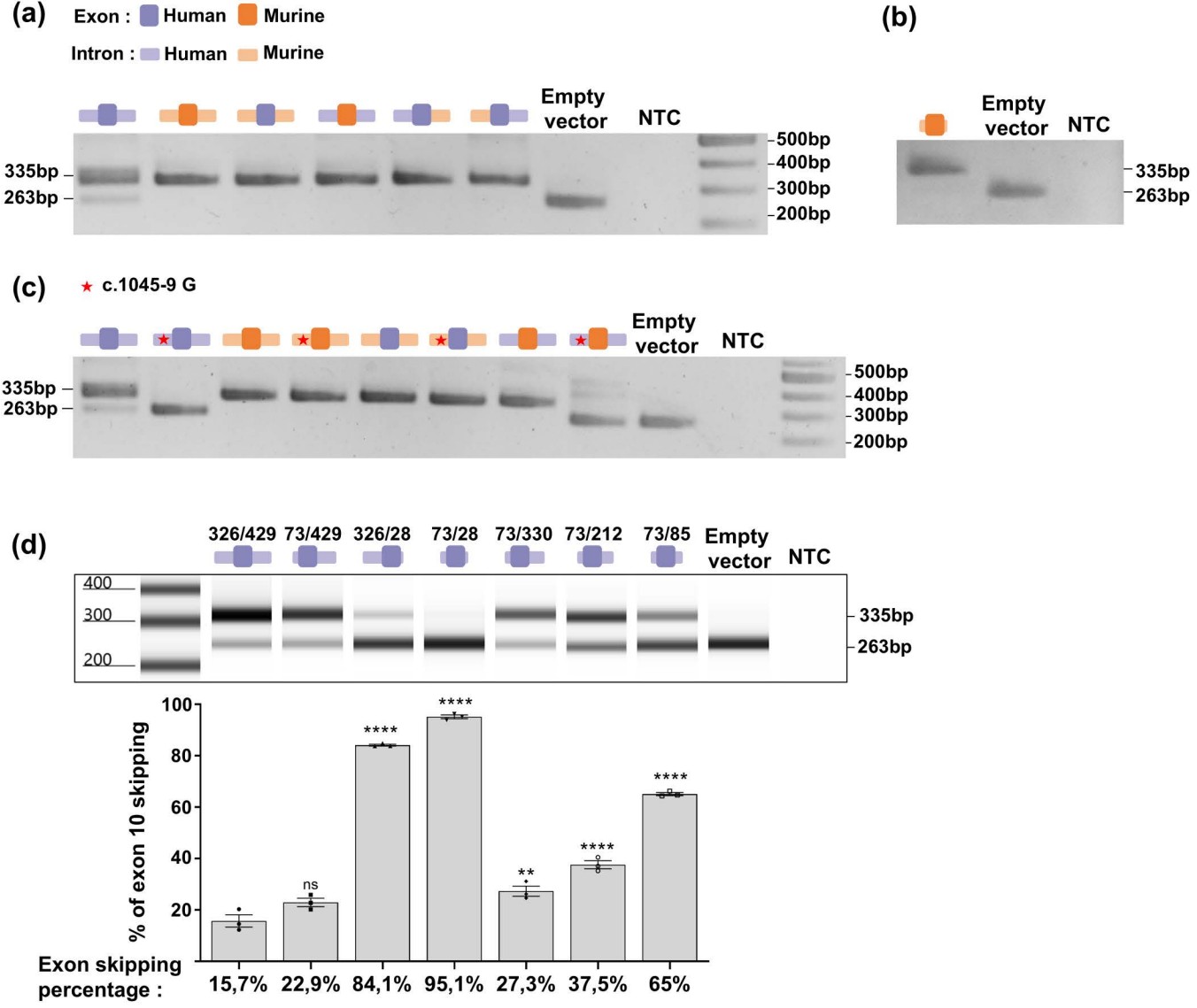

**Fig 3. Minigene assay profiling of human and mouse intronic sequences. (a)** Agarose gel electrophoresis of RT-PCR products from minigene assay on human, mouse and chimeric minigene constructs. Human (dark purple) or mouse (dark orange) exon 10 sequences were combined with only one or both human (light purple) or mouse (light orange) introns. **(b)** RT-PCR products of minigene assay using a minimal mouse minigene with 75 bp of intron 9 and 20 bp of intron 10. **(c)** Minigene assay assessing the effect of c.1045-9(G) (depicted with a red star), depending on the surrounding mouse or human sequences. **(d)** Representative RT-PCR products profile from minigene assay using human minigene with varying intron sizes. From left to right: (intron 9: 326 bp/intron 10: 429 bp); (73 bp/429 bp); (326 bp/28 bp); (73 bp/28 bp); (73 bp/330 bp); (73 bp/212 bp); (73 bp/85 bp) (upper panel). Relative quantification of exon 10 skipping after transfection with the minigene constructs above. Mean % of skipping is indicated below each histogram. Error bars represent SEM (n = 3). **** indicates p < 0.0001, ** indicates p < 0.01, ns indicates non-significant, one-way ANOVA followed by the Bonferroni post hoc multiple comparison test (lower panel). NTC: no-template control.

+9 to +20, the sequence is identical in all rodents (in which exon 10 is systematically included) but substantially variable among species vulnerable to exon 10 skipping.

In order to get more insight into the involvement of the intronic surrounding sequences, new human minigene constructs were designed with shortened *OCA2* intronic segments (Fig 3d). Similar to the murine construct (Fig

3b), the intron 9 sequence could be reduced to 73 bp (including the putative branch point at -47 bp) without significantly affecting the skipping ratio. This sequence contains the 40 bp stretch that is missing in the rodent sequence. By contrast, shortening of human intron 10 to a sequence of only 28 bp induced marked exon 10 skipping when combined with >300 bp of intron 9 and full skipping when both intron 9 and intron 10 sequences were shortened to 73 and 28 bp respectively. Additional constructs defined a minimal human minigene with preserved basal levels of skipping of less than 30%. This construct carried 73 bp of intron 9 and 330 bp of intron 10 (Fig 3d). This highlighted a major difference with the murine minimal sequence that requires only 20 bp of intron 10 to fully retain the exon in the minigene assay.

This study was extended by using antisense RNA oligonucleotide (ASO) mapping to investigate more precisely the role of different regions of exon 10 in retention/skipping control. Four ASO sequences of 22 nucleotides each were synthesized to cover the entire 72 bp exon 10 (Fig 4a). They were designed to hybridize the human sequence and were used to co-transfect HeLa cells with the human minigene at concentrations of 50 or 5 nM. At 50 nM, each of ASO1, ASO2 and ASO4 induced complete skipping, whereas ASO3 allowed a few percent of RT-PCR products to retain exon 10 (Fig 4b). The involvement of the four regions covered by the ASOs was confirmed using them at the lower concentration of 5nM (Fig 4c). ASO3 allowed retention of exon 10 in less than 30% of the transcripts whereas the 3 other ASOs still induced major skipping (~90%). We then used the same assay with the chimeric construct containing the human exon 10 flanked by the murine intronic sequences. Using this chimeric minigene, a concentration 50 nM of ASO (Fig 4d) was required to achieve a similar skipping profile to that observed with the human minigene sequence using 5nM ASO. Interestingly in the presence of murine intronic sequences and a concentration of 5nM ASO, none of the ASOs were able to substantially induce exon 10 skipping, which was detected in only a small and variable fraction of the RT-PCR products after treatment with ASO1, 2 and 4 (Fig 4e). These results confirmed that the balance of exon 10 retention/skipping relies on both the exon sequence and its intronic environment suggesting that co-evolution of these segments allowed basal levels of skipping in several groups of mammalian species but not rodents. This hypothesis is supported by the comparison of the nucleotide sequences across the whole region in species that do or do not express the skipped transcript (S2 Fig).

**The benign variant (c.1065G>A/p.Ala355=) rs1800404-T enhances exon 10 skipping in the minigene assay**

We next tested the most frequent exonic variant of *OCA2* that locates in exon 10. The synonymous c.1065G>A/p. Ala355=corresponds to the (T) allele of SNV rs1800404:C/T. It is highly predominant in European populations with an average allelic frequency of around 0.8, whereas it is less represented in Asian (f=~0.4) and African, African American populations (f=~0.2) as extracted from the GnomAd database [30]. Interestingly, statistical association studies using RNA-Seq data obtained on a collection of primary melanocytes highlighted a strong correlation between c.1065G>A) and exon 10 skipping [19]. As all of our patients with European ancestry carry c.1065G>A in *cis* to their *OCA2* exon 10 VUS, our minigene assay had so far been performed on a c.1065G>A background, which resulted in ~17% basal skipping levels in the WT control (Fig 1c). In order to monitor the potential impact of c.1065G>A on exon 10 skipping, a control minigene carrying c.1065G was compared with our standard control c.1065G>A minigene in the splicing assay. In addition, c.1080C>T/p.Ser360=previously shown pathogenic in *cis* to c.1065G>A [13] was tested in *cis* to either c.1065G or c.1065G>A. As shown in Fig 5, c.1065G allowed a significantly better retention of exon 10 (~4.5-fold) with basal skipping levels of only ~3%. The enhancement of skipping driven by c.1080C>T/p.Ser360=was significant in *cis* to c.1065G with an increase from 3% to 25% of skipped RT-PCR products. This level was however much weaker than in *cis* to c.1065G>A (65%). The additive effect of c.1065G>A was also observed in *cis* to the other previously studied rare synonymous variant c.1047C>T/p.Ile349= [13] (S5 Fig).

**Fig 4. ASO mapping assay of human *OCA2* exon 10 combined with human or mouse intronic sequences. (a)** Location of the human *OCA2* exon 10 sequences targeted by each of the four 22-base long ASOs. **(b-e)** For each condition, a representative screen tape electrophoresis is shown (upper) as well as the relative quantification of exon 10 skipping obtained after co-transfection with each ASO, mean % of skipping is indicated below each histogram (lower). Controls of the minigene assay are the result of transfection with the minigene only. NT: non-targeting oligonucleotide control, V: transfection vehicle control, NTC: no-template control. One-way ANOVA followed by the Bonferroni post hoc multiple comparison test, **** indicates $p < 0.0001$, ns indicates non-significant. Error bars represent SEM ($n = 3$). For multiple comparison, same letters indicate no significant differences, na: non-applicable ($n = 2$). **(b-c)** Human minigene with 50nM **(b)** or 5nM **(c)** ASOs. **(d-e)** Chimeric minigene with mouse introns and human exon with 50nM **(d)** or 5nM **(e)** ASOs.

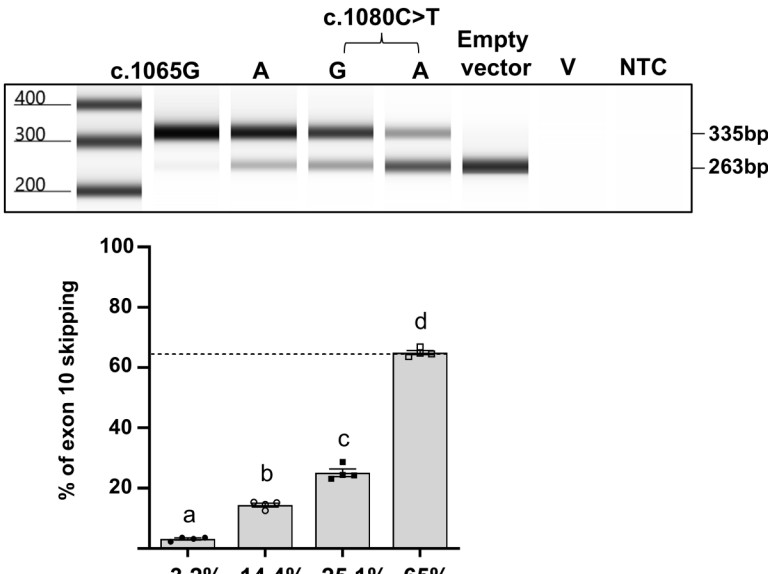

**Fig 5. Functional analysis of the effect of c.1065G>A on exon 10 skipping by minigene assay.** Representative screen tape electrophoresis of RT-PCR products from compared minigene assays of constructs with c.1065G or c.1065A. The additional effect of c.1080C>T/p.Ser360= is tested in *cis* to each of these variants. V: transfection vehicle control. NTC: no-template control. Relative quantification of exon skipping showing significant increase in skipping with c.1065A compared to c.1065G and the additional effect of c.1080C>T. Error bars represent SEM (n=4). A one-way ANOVA followed by the Bonferroni post hoc multiple comparison test was performed. Variables with different letters are significantly different (p<0.0001) (lower panel).

### rs1800404-T (c.1065G>A) is associated with lighter pigmentation in European population

The c.1065G>A variant significantly enhances skipping of exon 10, including when located in *cis* to a rare variant such as c.1080C>T, which likely contributes to reaching the threshold for pathogenicity. In the absence of a rare variant in *cis*, it induces a 4.5-fold skipping increase compared to c.1065G in the minigene assay. This raises the question of its functional impact on pigmentation in individuals without albinism, particularly in the European population for which this SNV (referred to as rs1800404-T in population genetic studies) has never been associated with pigmentation scores. Of note, rs1800404:C/T is located at ~5kb of rs1800407:C/T, of which the derived allele rs1800407-T (corresponding to c.1256G>A/p.R419Q) is reported as likely hypomorphic with suggested causative effects in lighter pigmentation [31]. In European populations, rs1800404 and rs1800407 are in significant linkage disequilibrium (LD, $r^2$: 0.27; D′: +0.95) as shown by analysing the LDlink database [32]. This pattern of LD generates 3 common haplotypes, rs1800404-T/rs1800407-C (f=0.783), rs1800404-C/rs1800407-C (f=0.141) and rs1800404-C/rs1800407-T (f=0.073), whereas haplotype rs1800404-T/rs1800407-T is much less represented (f=0.003). We hypothesized that the strong LD between these SNVs may hide the association of rs1800404:C/T to pigmentation scores in European populations with fair skin.

For this reason, we investigated the association of haplotypes derived from rs1800404 and rs1800407 to pigmentation phenotypes in a UK Biobank cohort with European reported ancestry (n=402871). Fig 6a, 6b shows the distribution of the different self-reported skin colours (6 categories) and hair colours (6 categories) in this population as well as of the genotypes at rs1800404 (f(T)= 0.792) and rs1800407 (f(T)= 0.086), illustrating the under-representation of haplotype rs1800407-T/rs1800404-T with only 2 double homozygotes (f=5 x $10^{-6}$) carrying the derived allele of each SNV (Fig 6c). In the selected cohort, the pattern of LD between rs1800404 and rs1800407 was consistent ($r^2$=0.33; D′=+ 0.96) with that reported in the European population of the LDlink database.

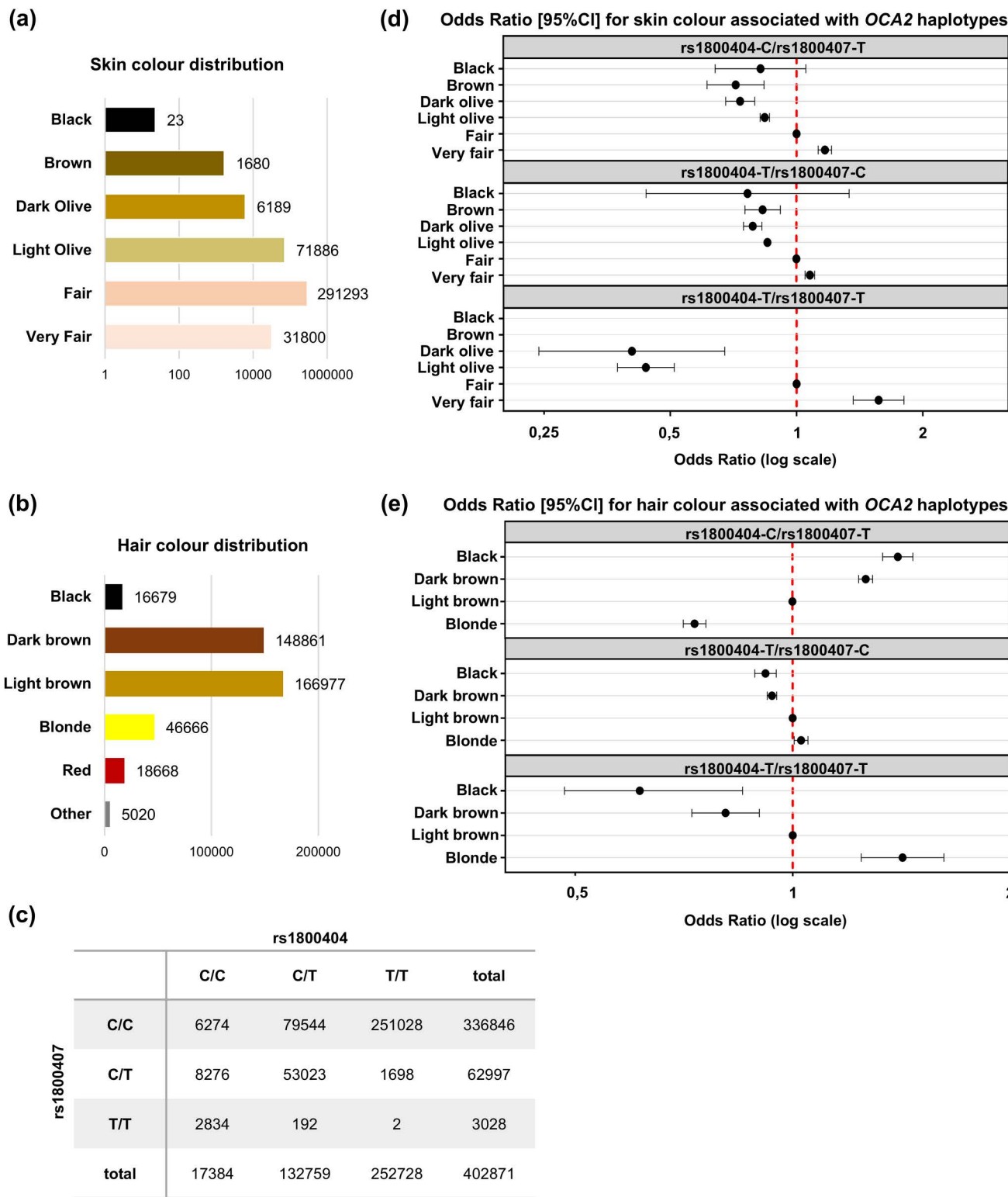

**Fig 6. Haplotypes derived from rs1800404:C/T and rs1800407:C/T to skin and hair colour in European subjects. (a-b)** Self-reported categorical skin **(a)** and hair **(b)** colour distribution in the studied UK Biobank population (n = 402871). **(c)** Distribution of the 9 possible genotypes for rs1800404:C/T and rs1800407:C/T in the studied population. Only 2 out of 402871 subjects are double homozygotes for the variants of each SNVs (f = $5.10^{-6}$). **(d-e)** Effects of rs1800404 and rs1800407 haplotypic combinations expressed as odd ratios. Each dot corresponds to the effects of *OCA2* haplotypes on skin/hair pigmentation considering the rs1800404:C/rs1800407:C haplotype as a reference. Major categories in the studied population (fair for skin, and light

brown for hair colour) are used as phenotypical references (red dashed vertical line). Haplotype effects were obtained from a multinomial regression model adjusted for age and sex. **(d)** All subjects were included, n = 402871, 6 categories. **(e)** Subjects with red hair or non-reported hair colour were excluded, n = 379183, 4 categories.

Taking the most represented phenotype as a reference, i.e., fair skin, and light brown hair colour, we then investigated the association of rs1800407-T and rs1800404-T alleles with skin and hair pigmentation using a multinomial regression model.

Under the assumption of proportional odds, single SNV analysis showed that the rs1800407-T allele was associated with lighter skin pigmentation (OR = 0.916 [0.901 - 0.932], p = 4.83 x $10^{-23}$) as was the rs1800404-T allele (OR = 0.924 [0.914 - 0.935], p = 1.12 x $10^{-38}$) (S6a Fig and S3a Table). In a joint model assuming additive effects of the two T alleles, these associations were significantly reinforced, OR = 0.798 [0.781 - 0.815] (p = 1.04 x $10^{-98}$) and OR = 0.846 [0.834 - 0.858] (p = 1.99 x $10^{-115}$), respectively. Detailed results of the haplotype analysis of these two SNVs are provided in S3a Table and are summarized in Fig 6d. Compared to the rs1800404-C/rs1800407-C haplotype, all three other haplotypes were associated with lighter skin pigmentation: rs1800404-T/rs1800407-C: OR = 0.853 [0.841 − 0.866] (p = 2.05 x $10^{-101}$); rs1800404-C/rs1800407-T: OR = 0.813 [0.796 − 0.831] (p = 1.10 x $10^{-78}$] and rs1800404-T/rs1800407-T: OR = 0.472 [0.426 − 0.523] (p = 2.56 x $10^{-46}$). This illustrates the strong correlation of rs1800404-T (c.1065G > A) to fair pigmentation, an association that can go undetected if LD between rs1800404 and rs1800407 is not taken into account. In turn, the impact of rs1800407-T on skin pigmentation tended to be significantly (p = 7.18 x $10^{-13}$) higher in *cis* to rs1800404-T allele (OR = 0.553) rather than to rs1800404-C allele (OR = 0.813).

Similar analyses were next conducted on hair pigmentation (Fig 6e, S3b Table; S6c, S6d Fig). The cohort was reduced to the four major hair colour categories (n = 379,183 subjects) excluding subjects with red or undefined hair colour. Single SNV analysis unexpectedly found that rs1800407-T was associated to dark hair colour rather than light (OR = 1.432[1.410 - 1.454] p < $10^{-306}$) whereas rs1800404-T was significantly correlated to fair hair (OR = 0.819 [0.810 - 0.828], p = 1.21 x $10^{-306}$), as expected under the assumption of playing a causal role. In a joint additive model, these opposite directions were confirmed with rs1800407-T associated with increased hair pigmentation (OR = 1.334 [1.310 − 1.359] (p = 4.35 x $10^{-205}$) and rs1800404-T allele associated with decreased hair pigmentation (OR = 0.919 [0.907 - 0.931], p = 8.00 x $10^{-39}$). Haplotype analysis further demonstrated that the rs1800404-T/rs1800407-C and rs1800404-T/rs1800407-T haplotypes were associated with lighter hair pigmentation (OR = 0.932 [0.92 - 0.944], p = 7.24 x $10^{-27}$) and OR = 0.692 [0.633 - 0.756], p = 3.44 x $10^{-16}$, respectively, compared to the rs1800404-C/rs1800407-C haplotype. By contrast, the rs1800404-C/rs1800407-T was markedly associated with darker hair pigmentation (OR = 1.376 [1.35 - 1.402], p = 7.99 x $10^{-236}$). These observations demonstrate that considering the haplotype rs1800404/rs1800407 is key in statistical association analyses with pigmentation-related phenotypes. Furthermore, these results support the hypothesis that the increase of exon 10 skipping triggered by rs1800404-T(c.1065G > A) negatively impacts pigmentation not only in the context of albinism but also in the whole population.

## The alternative *OCA2* transcript with skipped exon 10 may be expressed as a protein isoform with dominant-negative effect

We next asked how a moderate increase in skipping, such as that mediated by c.1065G > A, could significantly impact melanin production in the pigmented cells of healthy individuals. As exon 10 of *OCA2* is in frame with the rest of the coding sequence, skipping it theoretically results in the translation of an isoform lacking a stretch of 24 amino acids (from p.349 to p.372). If so, it raises questions about its potential function. In an attempt to predict the consequences of increased levels of the putative protein, which we will refer to as ΔE10-OCA2, we modelled and compared its topology to that of the native OCA2 protein initially published and revised by some of us [16,33]. This indicated a substantial change

in the arrangement of the first transmembrane helix domain that no longer packs with the main transmembrane helical bundle (Fig 7a). This prediction and the associated lower confidence scores (S7a Fig) indicate that ΔE10-OCA2 can fold with a likely lower structural stability than that of OCA2. Furthermore, multimer modelling predicts that the OCA2 protein can assemble as a homodimer [33]. This raises the question of a potential dominant-negative effect of increased levels of ΔE10-OCA2 resulting from increased skipping of exon 10. As shown in Fig 7b, multimer models of the ΔE10-OCA2 homodimer and the heterodimer with full-length OCA2 protein are expected with overall high confidence scores despite a reduced confidence at some predicted interface regions (S7b Fig). Overall, these models predict that the formation of heterodimers between full-length OCA2 and ΔE10-OCA2 isoforms is biologically feasible. Further experimental investigations will be required to functionally test the hypothesis of dominant-negative effect of ΔE10-OCA2 through heterodimerization.

## Discussion

In this study, we provide evidence that the human *OCA2* gene exhibits high sensitivity to sequence modifications within and surrounding exon 10, resulting in significant alterations of skipping levels that impact pigmentation in both pathological and physiological contexts. Previously, we classified two synonymous variants of exon 10 as pathogenic due to exon 10 skipping enhancement [13]. Here, we demonstrate that five out of six rare missense variants in exon 10 also increase exon skipping to varying degrees, which can contribute to the attainment of the pathogenic threshold. Functional assays combining murine and human sequences illustrate the involvement of the whole exon in exon retention/skipping control. A comparable sensitivity to sequence-dependent modulation of skipping levels has been reported and extensively studied in the case of *CFTR* exon 13 (historical name: exon 12) by Pagani's group [34–36]. Like *OCA2* exon 10 (72 bp), exon 13 of *CFTR* (87 bp) is in-frame with the upstream and downstream coding exons. Its skipping levels are strikingly dependent on multiple nucleotides all along the exon, with many single nucleotide variants (missense or synonymous) being pathogenic due to excessive skipping. Interestingly, the skipping level of *CFTR* exon 13 is much lower in mouse compared to human and non-human primates, which allowed the authors to identify critical exonic sequences involved in the splicing/skipping balance [35]. This raised hypotheses of dual selective pressures towards loss of exon retention and amino-acid sequence optimization in the course of evolution [36]. Although *OCA2* exon 10 has a similar profile to *CFTR* exon 13, i.e., basal levels of skipping in primates and no detectable skipped transcripts in rodents, our observations are not in favour of selection pressures towards codon or amino-acid optimization in species that do express basal levels of skipped transcripts such as reported for *CFTR* [35] and other genes [37]. Moreover, they highlight the involvement of the intronic surrounding sequences that have coevolved in mammalian species with basal levels of skipping. This leaves the possibility that the expression of the skipped transcript has been positively selected through evolution for a functional role in pigmentation control including within the human population like that proposed for other genetic variants of *OCA2* [38].

In line with this hypothesis, we show that the most frequent exonic human variant c.1065G>A (rs1800404-T), which is predominant in European populations with fair skin, significantly increases exon 10 skipping. This result functionally supports the strong association between c.1065G>A (rs1800404-T) and exon 10 skipped transcript previously reported after RNA-Seq analysis of biological samples [19]. In this same GWAS study that identified several loci associated with skin pigmentation in African populations, rs1800404-T displayed the highest probability of being causal. Strong association was also reported in several other populations with dark skin [20–23] but not in European populations with lighter skin. We hypothesized that this may be due to strong LD between rs1800404 and the previously identified nearby SNV rs1800407. Indeed, our haplotype association analysis of over 400,000 individuals with European ancestry shows that rs1800404-T, which causes exon 10 skipping enhancement, strongly correlates with lighter skin and hair in Europeans, whereas the direction of the effect of rs1800407 is not concordant with causality of hair colour (Figs 6e and S6c), as previously reported on a smaller cohort [39]. The association as well as the predominant frequency of rs1800404-T in the European population (f=~0.8) raise intriguing questions about its evolutionary selection such as its potential additional role in the adaptation of human skin colour in different regions of the globe [40].

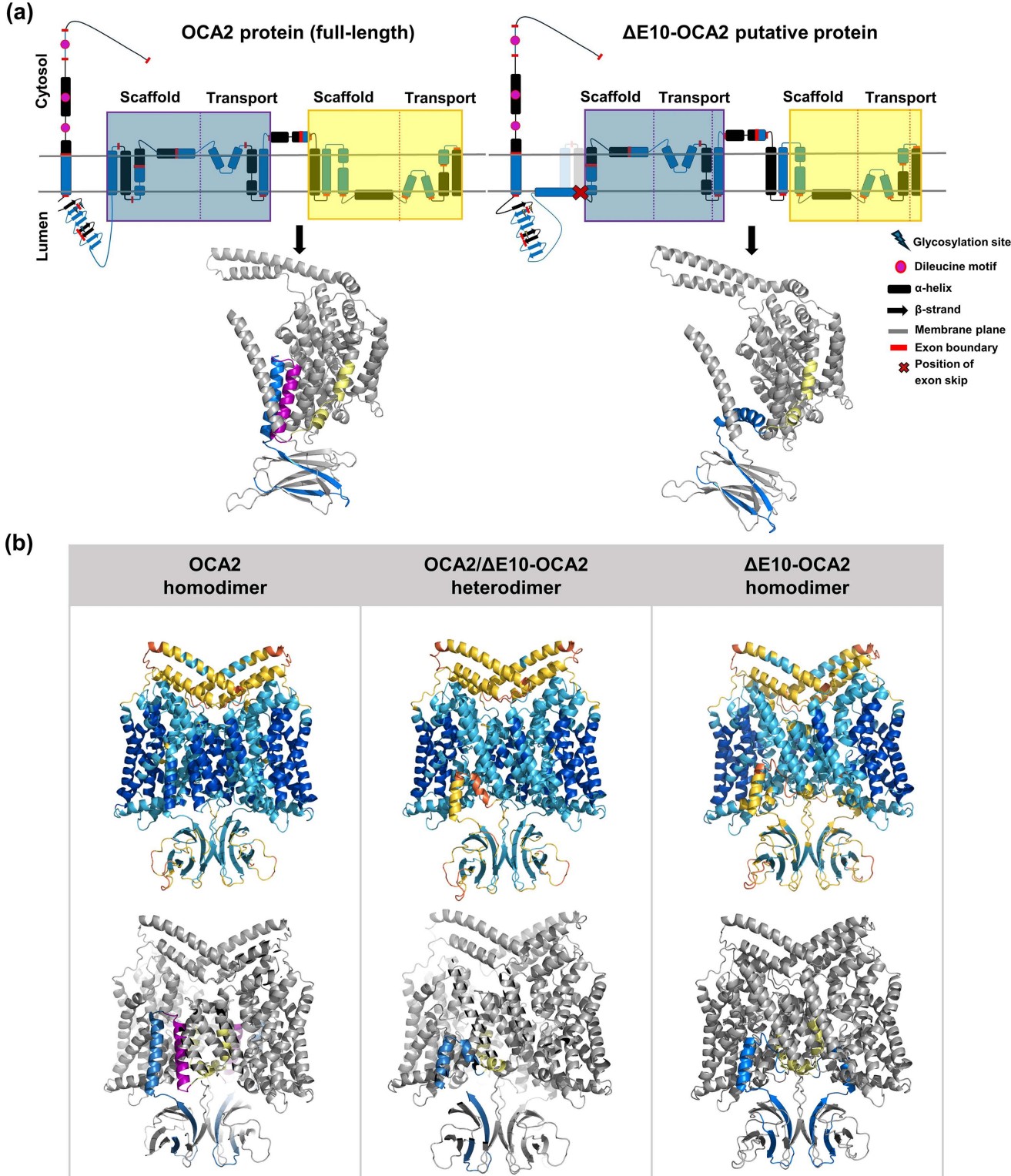

**Fig 7. Conformation prediction of the ΔE10-OCA2 protein compared to the native OCA2 protein. (a)** 2D transmembrane topology (upper panel) of the OCA2 protein (left) and the ΔE10-OCA2 protein (right) and corresponding structural models (lower panel) of monomers (ColabFold). The OCA2 protein topology is shown as transparent on the ΔE10-OCA2 topology. Boxes highlight the pseudo inverse repeat with scaffold and transport domains;

blue/black indicate alternating coding exons. Glycosylation sites correspond to N214, N218, N273; Dileucine motives are at position p.96-102, p.135-140, p.154-159. Structural models of the OCA2 and ΔE10-OCA2 monomers highlight exon 9 (blue), exon 10 (purple), and exon 11 (yellow) encoded domains. The cytosolic N-terminal 165 residue stretch with undefined conformation is not included in the structural models. **(b)** Structural models of three AlphaPulldown dimer predictions starting at residue 166. Each model is shown in cartoon representation with both chains visible. Upper panel: Models are coloured according to the pLDDT scale, where dark blue indicates high confidence (>90), light blue represents moderate confidence (70–90), yellow low confidence (50–70), and orange very low confidence (<50). Lower panel: same models highlighting exon encoded domains as in **(a)**.

Our results together with previously reported findings [19] suggest that *OCA2* exon 10 skipping levels induce proportional decreases in pigmentation in both pathological and physiological contexts. This prompts us to consider the underlying mechanisms that could enable such a fine-tuned phenotypic response. Although we have no evidence for the expression of the corresponding protein ΔE10-OCA2, its potential dominant-negative effect would explain the dose-sensitive response suggested by statistical associations in healthy populations. Given that the OCA2 protein may well function as a dimeric transporter and shares homology with members of the NACT family [33,41] which includes transporters with cooperative or interdependent subunit mechanisms, the presence of the non-functional ΔE10-OCA2 monomer within a heterodimer could potentially compromise overall chloride conductance, resulting in a dose-dependent down-regulation of melanogenesis. Alternatively, heterodimer misfolding could induce endoplasmic reticulum-associated degradation (ERAD) via the ubiquitin–proteasome system, similar to the quality control that is triggered by the well-studied ΔF508 form of the CF chloride conductance protein [42]. Thus, whether engaged in a dimer or as a structurally modified monomer, ΔE10-OCA2 could exert a dominant effect by interfering with *OCA2*-dependent Tyrosinase processing and/or activity that is key to melanogenesis activation [8].

What are the implications of the present study for molecular diagnosis of albinism? Oculocutaneous albinism is considered as a classical recessive rare disease caused by Mendelian rare variants with strong functional effects as opposed to common genetic diseases caused by multiple common variants with weak effects. We have recently provided several examples of complex determinism of albinism that challenge this paradigm [43–45]. Here, we show that rare variants of *OCA2* exon 10 such as c.1047C>T and c.1080C>T that do enhance skipping without inducing full exclusion, have a much weaker effect in *cis* to c.1065G compared to c.1065G>A, questioning the role of the haplotype in the refinement of pathogenicity threshold. All variants causing skipping enhancement that were identified in our previous report (c.1047C>T/p.Ile349=; c.1080C>T; p.Ser360=; c.1095_1103del/p.Ala366_368del) or in this study (c.1048G>A/p.Val350Met; c.1081C>G/p.Leu361Val) were in *cis* to c.1065G>A in the corresponding patients of our local cohort. Yet, their impact in *cis* to c.1065G is unproven. As indicated in the Results section (Table 1), the worldwide frequency of four of the missense variants is rather high with several homozygote individuals. Looking into more details, this is due to a significantly more elevated frequency in some non-European populations than worldwide (S4 Table). For instance c.1048G>A/p.Val350Met is significantly more frequent in South Asia (f = $1.3.10^{-3}$) than in the worldwide population including the UK Biobank subjects (f = $9.79.10^{-5}$). Two presumably non-albino homozygotes carrying this variant are scored in the South-Asian population. Their exome sequence confirms that none of the corresponding alleles have the c.1065G>A (f = 0.39 in South Asia) in *cis* to c.1048G>A/p.Val350Met (S8 Fig). The same observation can be made with the three other missense variants shown here to significantly increase skipping in *cis* to c.1065G>A (c.1064C>T/p.Ala355Val; c.1109T>C/p.Ile370Val; c.1114G>A/p.Asp372Asn, Fig 1c). Sequencing data indicate that they systematically segregate with c.1065G in presumably healthy homozygotes (S8 Fig) leaving the possibility that the pathogenic threshold is reached only when they lie in *cis* to c.1065G>A. In the context of future diagnosis, these observations suggest that any such variant may impact pigmentation within the physiological range if located in *cis* to c.1065G and be pathogenic in *cis* to 1065A. Not only c.1065 (rs1800404) but the entire haplotype that regulates *OCA2* expression would be worth analyzing to enable the prediction of *cis*-interfering or *cis*-additive effects of common variants. This includes the enhancer of *OCA2* within *HERC2* and its causal polymorphism rs12913832 [5]. As a result, the identification of haplotypes with combination

of SNVs that result in high skipping ratio may resolve the genetic determinism of some of the patients with so far only one single identified mendelian variant in *OCA2*. In the light of the new findings presented here, several already documented variants with inconclusive or conflicting interpretation would be worth reconsidering. For instance, the synonymous variant c.1113C>T/p.Gly371= (rs1800405-A) is reported as likely causal in a family of Vietnamese patients [46] whereas it is considered as benign due to its frequency in the range of the percent worldwide reaching ~10% in Admixed American populations with as many as 1% of homozygotes (GnomAd database). Of note, linkage disequilibrium with c.1065 (rs1800404) makes the haplotype rs1800405-A/rs1800404-T absent from the 5008 haplotypes included in the LDlink database. Testing of the 4 possible haplotypes in the minigene assay together with haplotype identification in the corresponding patients should allow to resolve their genetic diagnosis. This strategy applies to other variants listed in S4 Table.

As summarized in Fig 8, this study strongly suggests that c.1065G>A (rs1800404-T) contributes to paleness/hypopigmentation on the whole spectrum from a healthy population with dark skin to patients with albinism. Including c.1065G>A

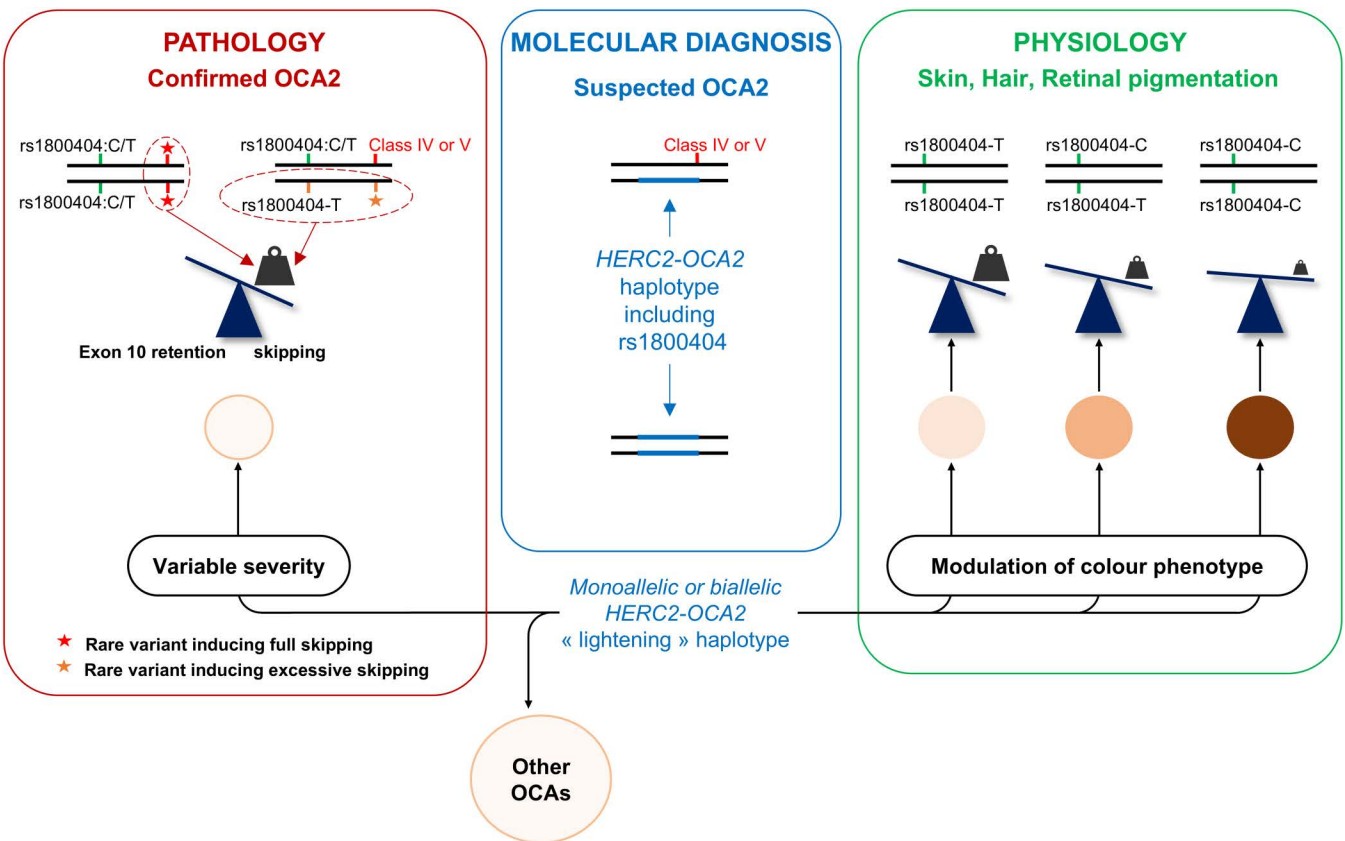

**Fig 8. *OCA2* exon 10 skipping and pigmentation across a continuous spectrum from pathology to physiology.** In the context of OCA2 diagnosis (red box), rare variants such as c.1045-15T>G (red star) result in full skipping of *OCA2* exon 10 whether in *cis* to c.1065G [53] or c.1065G>A [13]. Rare variants such as c.1080C>T that enhance skipping levels leaving some transcripts that retain exon 10 (orange star) are pathogenic when in *cis* to c.1065G>A (rs1800404-T). This frequent variant increases basal levels of exon 10 skipping on its own and is benignly associated to lightening of the skin and hair in healthy individuals of all ethnicities including with European ancestries (this study). The effect of rs1800404-T is dose-dependent as homozygotes have a significantly lighter skin and hair than heterozygotes (green box). The retinal pigmentation is likely affected given the contribution of rs1800404-T (c.1065G>A) to the ocular pathology in patients with OCA. Patients with only one identified pathogenic variant should be investigated for haplotypic combinations that would increase skipping in the transcripts encoded by the second allele (blue box). The probability of such a pathogenic haplotype will depend on the patient's ethnicity (for instance c.1114G>A/Asp372Asn is significantly more frequent in South Asia). A *HERC2-OCA2* "lightening" haplotype (in favour of increased exon 10 skipping ratio) is expected to contribute to modulation of both physiological pigmentation and severity of OCA2 depending on the second allele. Given the extreme sensitivity of the system to skipping levels, it can be expected that such a haplotype will influence the severity of other OCAs such as OCA1 driven by hypomorphic variants of *TYR*.

in genetic testing not only applies to OCA2 patients with missing heritability but also to patients with other types of albinism taking into account epistasis of *OCA2* on downstream genes. Whether this polymorphism is also associated with specific retinal phenotypes in albino and non-albino individuals merits future investigations as these findings could have direct implications for refining genetic diagnosis of albinism.

## Materials and methods

### Ethics statement

This study was approved by our local ethics committee: Comité de Protection des Personnes Bordeaux Outre-Mer III. Written informed consent was received from the patients and the participating families. The UK Biobank has received approval from the National Information Governance Board for Health and Social Care and the National Health Service North West Centre for Research Ethics Committee (Ref: 11/NW/0382). All investigations were conducted in accordance with the tenets of the Declaration of Helsinki.

### Patient cohort

The local cohort consists of over 3,000 patients with clinical signs of albinism, such as hypopigmentation of the skin, hair and eyes, nystagmus, foveal hypoplasia and iris translucency. These patients have been assessed for variants in albinism genes using a previously described gene panel until 2022 [12], and since then using the panel described in [16].

### Recombinant plasmid constructs and site-directed mutagenesis

The minigene containing the human wild-type (WT) *OCA2* exon 10 in pSPL3B vector has been described [13]. For constructing the minigene with the murine *Oca2* sequences, genomic DNA was extracted from C57BL/6J mouse tail-tip using "Phire Tissue Direct PCR Master Mix" (Thermo Fisher Scientific) following manufacturer's recommendations. Exon 10 flanked by 335 bp of intron 9 and 313 bp of intron 10 were amplified with "Q5 High Fidelity PCR Kit" (New England Biolabs) according to manufacturer's instructions and cloned in pSPL3B vector by homologous recombination as previously described [13]. PCR product was treated with the "In-Fusion HD Cloning kit with Cloning Enhancer Treatment" (Takara Bio) and recombined with Not I linearized pSPL3B vector. Both human and mouse minigenes served as a template to generate constructs with shortened introns. For the human minigene constructs, sequence of intron 9 was reduced to 73 bp that include the putative branch point. Four versions with different amounts of intron 10 sequences were cloned (330 bp, 212 bp, 85 bp or 28 bp) as well as combinations of shortened intron 9 (73 bp) and intron 10 (429 bp) or intron 9 (326 bp) and intron 10 (28 bp). The mouse minimal minigene was shortened to 75 bp for intron 9 and 20 bp for intron 10. All chimeric human/mouse constructs containing exon 10 flanked by around ~400 bp of intron 9 and 10 were synthesized by Eurofins–Genestrand. Lyophilized DNA was resuspended in nuclease-free water, cloned in pMiniT vector using the "PCR cloning kit" (New England Biolabs). The inserts were amplified as described above. PCR products were extracted from 1% agarose gel and purified using the "NucleoSpin Gel and PCR Clean-up kit" (Macherey-Nagel) and cloned in pSPL3B vector as above. All constructs were confirmed by Sanger sequencing (Eurofins). All primers that were used in this study are listed in S1 Table. For site-directed mutagenesis, single nucleotide variants of interest were introduced into the reference sequence of the minigene with either standard "QuickChange Site-Directed Mutagenesis Kit" or with optimized "QuickChange Lightning Site-Directed Mutagenesis Kit" (Agilent) following the manufacturer's recommendations. Mutagenesis primers were designed as recommended and are listed in S1 Table. Targeted mutagenesis was confirmed by Sanger sequencing (Eurofins).

### RT-PCR on RNA from cultured cells and tissues

All animal tissue samples used in this study were acquired from available post-mortem tissues from other studies involving wild-type control animals (mouse and Fat Sand Rat) approved by Bordeaux University Ethics Committee and Comité

d'éthique en matière d'expérimentation animale de Strasbourg (CREMEAS). Total RNAs from one-month C57BL6/J mouse skin, as well as skin and optic cups of Fat Sand Rat *Psammomys obesus*, were purified with "RNeasy Mini kit" (Qiagen). Total RNAs from human primary melanocyte culture were obtained as described [13]. First-strand cDNA synthesis was carried out with 1μg of total RNAs by oligo(T)-primed reverse-transcription using M-MLV reverse transcriptase (Promega) following the manufacturer's instructions. The resulting cDNAs were amplified by PCR with "Q5 High Fidelity PCR Kit" (New England Biolabs) containing human (forward: 5′ CTCCGCGGAAGTGTAGAAAC 3′ and reverse: 5′ GTGAAGAGGAGCATGGTGGT 3′) or mouse (forward: 5′ CTCTCCAACAGACTCGCCTC 3′ and reverse: 5′ AGGAGCATCGTAGTGACATTGT 3′) primers located in exon 9 and 13, or *Psammomys obesus* primers located in exon 8 and 12 (forward: 5′ CCTCCAACAGACCCACCTTG 3′; reverse: 5′ GTGGTGACATTGTCTAAGAAGGC 3′). PCR products were separated by electrophoresis through 1.5% agarose gel, verified by Sanger sequencing (Eurofins) and aligned with reference *OCA2* transcripts sequences (human: ENST00000354638.8, mouse: ENSMUST00000032633.12, www.ensembl.org), (*Psammomys obesus*: XM_055607524.1, www.ncbi.nlm.nih.gov).

### Antisense RNA oligonucleotide (ASO) synthesis

ASOs were designed to map on human *OCA2* exon 10 with the following sequences: ASO1: 5′-CUGCCAGAGUUCUGUGCACGAU-3′; ASO2: 5′-GCAAGGGAACCCAGCAUGGCUG-3′; ASO3: 5′-CCAGUGCUG CCAGUGCUGCAAG-3′; ASO4: 5′-AUCGCCAAUCACAGCCAGUGCU-3′. A non-targeting RNA oligonucleotide was used as a negative control (5′- AGUUCACACUGGAAUCAAAGGU-3′). ASOs were synthesized on a micromolar scale by the β-phosphoramidite method with backbone modifications (phosphothioate and 2′-methoxyethyl) using an H8 automated synthesizer (K&A Labs, Germany). Deprotection of the ASOs was performed according to the supplier's protocol. The concentrated crude ASOs were then resuspended in water. The sample concentration was determined from the absorbance at 260 nm and molar extinction coefficient of the ASO. This value was calculated using the Integrated DNA Technology online oligo analyzer tool, which uses the standard nearest neighbor method.

### Cell culture and transfection

HeLa cells were cultured in DMEM+GlutaMAX (Gibco) supplemented with 1% penicillin and streptomycin and 10% fetal bovine serum (Gibco). The day before transfection HeLa cells were seeded in 6-well plates with 150 000 cells per well. For minigene assays, transfections of HeLa cells were performed using FuGENE HD Transfection Reagent (Promega) with 3μg of plasmid DNA in a 1:1 ratio according to the manufacturer's recommendations. Cells were collected forty-eight hours after transfection by scraping. For ASO mapping experiment, HeLa cells were transfected with Lipofectamine3000 (Thermo Fisher Scientific) with 1μg of plasmid DNA and 5 or 50nM of ASO. Optimization of the standard protocol provided by the supplier consisted in increasing the mix incubation time from 5 to 20 minutes. To improve cell transfection efficiency, the transfection mix was applied to the cells in Opti-MEM 1X (Gibco). The cells were left in Opti-MEM at 37°C for 4 hours after which the Opti-MEM was replaced by complete culture medium. Cells were collected the day after transfection.

### Minigene products analysis

Total RNAs were purified using the "RNeasy Mini kit" (Qiagen). First-strand cDNA synthesis was carried out as described above. The resulting cDNA was amplified by PCR using SD6 and SA2 vector specific primers (SD6: 5′ TCTGAGTCACCTGGACAACC 3′; SA2: 5′ ATCTCAGTGGTATTTGTGAGC 3′) as described before [13]. PCR products were separated by electrophoresis through 2% agarose gel. In order to quantify *OCA2* exon 10 skipping, PCR products were first purified according to "NucleoSpin Gel and PCR Clean-up" (Macherey-Nagel) instructions and separated on a D1000 DNA Screentape system with TapeStation 4150 instrument (Agilent) following manufacturer's recommendations. The percentage integrated area (PIA) of the peak corresponding to the RT-PCR product without exon 10 is directly used to estimate the % of exon skipping.

### UK Biobank cohort

The UK Biobank is a large well-studied prospective cohort with more than 500,000 participants recruited from across the UK. Participants completed a baseline questionnaire including self-reporting of their skin and hair colour [47]. Participants provided extensive health, lifestyle, and genetic data, including self-reported traits such as skin and hair color, captured during baseline assessment visits. Genotyping was performed using custom UK Biobank Axiom arrays, and genotype imputation was conducted centrally using the Haplotype Reference Consortium and UK10K reference panels. Variant-level data were extracted from the imputed dataset using PLINK v2.0 by specifying genomic coordinates or rsIDs of interest. Relevant pigmentation-related traits were extracted using the DNA Nexus cohort browser and Table Exporter v2.0.82.

### OCA2 protein model building and dimer prediction

OCA2 monomer structures were generated using a local installation of ColabFold [48]. Dimeric models were constructed using the AlphaPulldown pipeline [49], which incorporates the AlphaFold-Multimer framework [50,51]. For both monomeric and dimeric predictions, multiple sequence alignments were obtained using MMseqs2 [52]. For each of the three dimer configurations, five structural models were generated, and the top-ranked model based on the internal scoring metric was selected for downstream analysis.

### Statistical analysis

Functional assays: all the results are expressed as mean value±standard error of the mean (SEM) of at least 3 independent experiments, unless stated otherwise. Statistical analysis was performed with GraphPad Prism version 8.0.2 for Windows using one-way ANOVA followed by the Bonferroni post hoc multiple comparison test.

Population genetics analysis: *OCA2* haplotypes derived from rs1800404:C/T and rs1800407:C/T were inferred using the *haplo.stats* R package. Associations between OCA2 SNVs/haplotypes and skin and hair pigmentation were analyzed using polytomous regression models (including both multinomial and proportional odds models), adjusting for age and sex. The most common phenotype was used as the reference category: fair skin and light brown hair, respectively. Regression analyses were conducted using the *vglm* function from the *VGAM* R package. Haplotypes with estimated frequencies below 0.001 in the UK Biobank data were excluded from statistical association analyses.

## Supporting information

**S1 Fig. Robust quantification of skipping levels by minigene assay using DNA Screentape analysis.** (a) Whole gel image of RT-PCR products corresponding to Fig 1b using D1000 DNA Screentape system on an Agilent TapeStation 4150 instrument. PCR products derived from minigene transcripts are detected through fluorescence around 300 bp. The upper (1500 bp) and lower (25 bp) markers, in D1000 loading dye, are size markers used as references for assessing the DNA fragments sizes in a sample. NTC=no-template control. (b) Electropherogram obtained following electrophoresis of RT-PCR products from the control minigene, shown in the gel above. The percentage integrated area (PIA) of the peak corresponding to the RT-PCR product without exon 10 is directly used to estimate the % of exon skipping as illustrated in the table below.
(TIF)

**S2 Fig. Alignment of *OCA2* sequences in and around exon 10 (or its equivalent) from different species that do or do not express the transcript without exon 10 (or its equivalent).** (a) Alignment of *OCA2* intron 9 sequences. Sequences alignment was performed on Clustal Omega (https://www.ebi.ac.uk/jdispatcher/msa/clustalo). Conserved nucleotides are identified by * while non homologous nucleotides are indicated by -. The human sequence is hightlighted in yellow and the putative splicing branch point is in red, 47 bp upstream the intron 3′ splice site (note that for mouse, the

branch point is predicted at -37 bp, not shown on the alignment). Intronic nucleotides near the exon highlighted in blue indicate highly conserved sequences across mammalian species. (b) Alignment of *OCA2* exon 10 sequences. Codon positions are indicated above the alignment. Non conserved nucleotides are highlighted: green for T, pink for C, blue for G, red for A. (c) Alignment *OCA2* intron 10, legend as for (a).
(TIF)

**S3 Fig. Experimental results of minigene assays resulting from nucleotide swapping between human and mouse exon 10 sequences supporting** Fig 2d**.** (a) Products of RT-PCR performed on mouse skin or optic cup RNAs using primers in exon 9 and 13 (see Methods) (b-c) Agarose gel electrophoresis of RT-PCR products of human (b) and mouse (c) minigene assay showing the effect of nucleotide swapping between human and mouse exon 10 sequences. Sequences of human origin are indicated in purple; sequences of murine origin, in red. NTC = no-template control.
(TIF)

**S4 Fig. Transcript profile of *OCA2* in *Psammomys obesus* (Fat Sand Rat).** (a) Alignment between *OCA2* exon 10 and surrounding sequences of human, mouse and *Psammomys obesus* (human: ENSG00000104044; mouse: ENS-MUSG00000030450; *Psammomys*: NW 026590486.1 (1302928–1626514)). Sequence alignment was performed on Clustal Omega (https://www.ebi.ac.uk/jdispatcher/msa/clustalo). Upper case letters refer to exon sequence (also highlighted in yellow) and lower case letters to intronic sequences flanking the exon. Intronic nucleotides highlighted in blue indicate similar sequences between the species, red indicates homology between mouse and *Psammomys* but not human whereas pink indicates nucleotides that are specific to one of the 2 rodent species. (b), Agarose gel electrophoresis of RT-PCR products of *Oca2* from P*sammomys* skin and eye total RNAs (primers in exon 8 and 12, see S1 Table). RT-PCR products correspond to the transcript correctly spliced with all exons (414 bp). This experiment was done on the skin of three different animals numbered 1 to 3, and on one eye of animal n°1. NTC = no-template control.
(TIF)

**S5 Fig. Cumulative effect of synonymous c.1047C > T/p.Ile349= and c.1065G > A/p.Ala355=.** Minigene assay evaluated by conventional agarose electrophoresis showing the effect of c.1047C > T/p.Ile349 = equivalent to the increase in skipping induced by c.1080C > T that was precisely quantified by Screen tape analysis (Fig 5). Variant c.1047C > T is comparatively tested in *cis* to c.1065G or c.1065A as indicated. NTC: no-template control.
(TIF)

**S6 Fig. Skin or hair colour categories association to genotypes at rs1800407 and rs1800404 in the UK Biobank population with European ancestry.** (a-b) Comparative association of rs1800407:C/T and rs1800404:C/T to skin colour. (c-d) Comparative association of rs1800407:C/T and rs1800404:C/T to hair colour. Association of skin colour (a) or hair colour (c) to the 3 genotypes (C/C, C/T, T/T) at rs1800407 (left) or rs1800404 (right) in the UK Biobank with European ancestry (n = 402871). Categories are indicated by corresponding colours. For each SNV analyzed separately, significant association is detected between the (T) allele and paleness of the skin (a). However, for the hair colour (c), the effect of rs1800407:C/T is opposite to what is expected, with (T) associated to darker rather than lighter hair colour. The effect of rs1800404-T, on the other hand, is in agreement with a contribution to paleness of both skin and hair (see text for statistical values). (b-d) Association of skin colour (b) or hair colour (d) to the 9 combinations of genotypes at rs1800404:C/T and rs1800407:C/T. Double heterozygotes with *cis* or *trans* segregation of the variants cannot be discriminated. This illustrates the strong association of rs1800404-T with hypopigmentation of the skin and hair independently of rs1800407-T as statistically shown in Fig 6.
(TIF)

**S7 Fig. Compared statistical analysis for conformation and dimerization prediction of the native OCA2 protein and the ΔE10-OCA2 protein.** (a) Structural models of two ColabFold OCA2 monomers coloured by per-residue

confidence (pLDDT). Models are truncated removing the first 165 residues representing the N-terminal disordered region. Models are coloured according to the pLDDT scale, where dark blue indicates high confidence (>90), light blue represents moderate confidence (70–90), yellow low confidence (50–70), and orange very low confidence (<50). Each model is shown in cartoon representation with both chains visible. Below: Predicted Aligned Error (PAE) maps the two monomer models. PAE maps visualise the predicted alignment error (in Angstroms) between residue pairs across and within chains. Blue regions indicate high confidence (low error), and red regions represent low confidence (high error), with values capped at 30 Å. Diagonal blocks correspond to intrachain predictions; off-diagonal blocks indicate inter-chain confidence. (b) PAE maps for three AlphaPulldown dimeric models. Same as above. Below: Comparative summary of PAE maps for the three dimer AlphaPulldown models. Each model is evaluated for intrachain confidence and interchain interaction quality based on PAE colour distribution. Intrachain confidence reflects the predicted accuracy of the individual monomer folds, while interchain confidence and interface clarity indicate the reliability and extent of the predicted dimer interface.
(TIF)

**S8 Fig. Haplotypic combinations involving a rare missense variant of exon 10 and c.1065G>A; rs1800404:C/T in GnomAd.** Sequencing data (GnomAd V4.0.1) from subjects carrying a rare missense or synonymous variant of exon 10 was examined for segregation with c.1065G>A. Number of homozygotes for each rare variant were: p.V350M (2 homozygotes); p.L354P (no homozygote); p.A355V (4 homozygotes); p.L361V (no homozygote); p.A368V (1 homozygote); p. I370T (9 homozygotes); p.G371= (479 homozygotes); p.D372N (1 homozygote). On all recovered sequences (n = 156), half were incomplete such that the haplotype could not be defined. On the remaining: all presumably healthy homozygotes carried the rare variant segregating in *cis* to c.1065(G), n = 6: 2 with p.V350M and 4 with p.A355V (see corresponding sequence panels). As for the heterozygous carriers, only 3 heterozygotes for the same variant (D372N) carried the variant in *cis* to c.1065G>A (last panel). Two were double heterozygotes and one was homozygote for c.1065G>A. All in all, no presumably healthy homozygote for a rare missense variant that enhances exon 10 skipping carried the variant in cis to c.1065G>A in agreement with the benign nature of such haplotype. The position of c.1065 is indicated by a blue arrow, the position of the rare variant by a red arrow.
(TIF)

**S1 Table. Oligonucleotidic sequences of all primers used for mutagenesis, cloning or RT-PCR reactions.**
(PDF)

**S2 Table. OCA2 transcript identification for the species that were included in this study.** Informations were collected on Ensembl: http://www.ensembl.org/index.html except for *Psammomys obesus* informations were obtained on NCBI: https://www.ncbi.nlm.nih.gov/. Abbreviations: Δ10 = exon 10 skipped.
(PDF)

**S3 Table. Statistical data for rs1800404 and rs1800407 haplotypic association studies performed on the UK Biobank population with European ancestry.** (a) Association to skin colour. (b) Association to hair colour.
(PDF)

**S4 Table. Allelic frequencies of selected SNVs of exon 10 illustrating high degree of variability across the geographical subpopulations.** Allele frequencies provided in gnomAD v4.1.0 (including the UK Biobank). For each variant, the population with the highest frequency is highlighted in red, the one with the lowest frequency in green. NA: non applicable (too many citations).
(PDF)

## Acknowledgments

The authors would like to thank Genespoir, the French albinism association, for its invaluable support and the significant help it provides to patients.

## Author contributions

**Conceptualization:** Panagiotis I. Sergouniotis, Benoit Arveiler, Sophie Javerzat.

**Formal analysis:** Elina Mercier, David-Alexandre Trégouët, Shahram Mesdaghi, Daniel J. Rigden, Sophie Javerzat.

**Funding acquisition:** Benoit Arveiler, Sophie Javerzat.

**Investigation:** Elina Mercier, David-Alexandre Trégouët, Vincent Michaud, David J. Green, Shahram Mesdaghi, Orane Le Gallais, Orphée Jouannigot-Castano, Sophie Javerzat.

**Methodology:** Elina Mercier, David-Alexandre Trégouët, Sébastien Campagne, David J. Green, Shahram Mesdaghi, Catherine Estay-Ahumada.

**Project administration:** Benoit Arveiler, Sophie Javerzat.

**Resources:** Vincent Michaud, Daniel J. Rigden, David Hicks, Panagiotis I. Sergouniotis.

**Supervision:** Sophie Javerzat.

**Validation:** Sophie Javerzat.

**Visualization:** Elina Mercier, David-Alexandre Trégouët, Sébastien Campagne, Shahram Mesdaghi, Sophie Javerzat.

**Writing – original draft:** Elina Mercier, Sophie Javerzat.

**Writing – review & editing:** Elina Mercier, Vincent Michaud, Benoit Arveiler, Sophie Javerzat.

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
