## [Decision Letter · Decision Letter 0]

6 Aug 2025

PGENETICS-D-25-00764

From paleness to albinism: Contribution of OCA2 exon 10 skipping to hypopigmentation

PLOS Genetics

Dear Dr. Javerzat,

Thank you for submitting your manuscript to PLOS Genetics. The manuscript has been evaluated by two external reviewers. As you will see, their comments are both positive with regard to quality of the work and potential impact. However, there are a number of minor concerns that we ask you address in a minor revision. 

Please submit your revised manuscript within 30 days Sep 05 2025 11:59PM. If you will need more time than this to complete your revisions, please reply to this message or contact the journal office at plosgenetics@plos.org. Please include the following items when submitting your revised manuscript:

We look forward to receiving your revised manuscript.

Kind regards,

Gregory S. Barsh

Academic Editor

PLOS Genetics

Santhosh Girirajan

Section Editor

PLOS Genetics

Aimée Dudley

Editor-in-Chief

PLOS Genetics

Anne Goriely

Editor-in-Chief

PLOS Genetics

**Journal Requirements:**

At this stage, the following Authors/Authors require contributions: Elina Mercier, David-Alexandre Trégouët, Sébastien Campagne, Vincent Michaud, David J Green, Shahram Mesdaghi, Orane Le Gallais, Orphée Jouannigot-Castano, Daniel J Rigden, Catherine Estay-Ahumada, David Hicks, Panagiotis I Sergouniotis, Benoit Arveiler, and Sophie Javerzat. Please ensure that the full contributions of each author are acknowledged in the "Add/Edit/Remove Authors" section of our submission form.

The list of CRediT author contributions may be found here: https://journals.plos.org/plosgenetics/s/authorship#loc-author-contributions

https://journals.plos.org/plosgenetics/s/submission-guidelines#loc-parts-of-a-submission

- ® on page: 23

- TM on page: 23.

4) We note that your Data Availability Statement is currently as follows: "Novel variants have been uploaded to ClinVar public database.". Please confirm at this time whether or not your submission contains all raw data required to replicate the results of your study. Authors must share the “minimal data set” for their submission. PLOS defines the minimal data set to consist of the data required to replicate all study findings reported in the article, as well as related metadata and methods (https://journals.plos.org/plosone/s/data-availability#loc-minimal-data-set-definition).

2) If any authors received a salary from any of your funders, please state which authors and which funders..

6) Please ensure that the funders and grant numbers match between the Financial Disclosure field and the Funding Information tab in your submission form. Note that the funders must be provided in the same order in both places as well. Currently, the Financial Disclosure states there was no funding received.

**Reviewers' comments:**

Reviewer's Responses to Questions

**Comments to the Authors:**

Reviewer #1: The authors of “From paleness to albinism: Contribution of OCA2 exon 10 skipping to hypopigmentation,” present an interesting investigation of a collection of variants with a range of population frequency and their effect on alternative splicing of OCA2 exon 10 and pigmentation phenotype. I believe this work is important to the fields of population genetics, pigmentation biology, and clinical genetics. The breadth of this work is impressive, including variant analysis, functional assays of splicing, evolutionary alignments, phenotype association, and modeling molecular consequences on the protein level. I believe this is a worthy study, but I do suggest the following revisions:

Major issues:

1. Computational predictors of splice site disruption (i.e. SpliceAI and Pangolin) are low for the variants described in Table 1 and NM_000275.3:c.1080C>T, the previously described p.Ser360= variant. The authors should explore other biological explanation of these variants’ obvious effect on exon 10 skipping. It would be worthwhile to explore whether these variants fall within eCLIP-seq peaks in data available through publicly available databases. No additional experiments are needed, but RNA-protein interaction with splicing factors may explain this phenomenon better than splice-site disruption and would be a worthwhile investigation.

2. Along these same lines, NM_000275.3(OCA2):c.1045-9T>G, discussed in “Vulnerability of human OCA2 to exon 10 skipping relies on the combination of exonic and intronic sequences” and Figure 3c does have significant SpliceAI scores 0.24 (acceptor loss -9bp (the canonical acceptor site)) 0.22 (acceptor gain, +54bp), and Pangolin score (Splice Loss 0.47, -9bp). Does this variant cause exon 10 skipping or does it create an additional splicing product with partial retention of the 5’ end of exon 10 (i.e. new splice site has now moved +45bp into exon 10), as the SpliceAI scores predict? Is there evidence or literature of exon 10 skipping vs. exon 10 missplicing for this variant?

Minor Issues

3. Lines 115-116: The authors state that the probands with the c. c.1048G>A/p.Val350Met and c.1081C>G variant are reported to have an additional deep intronic OCA2 variants in trans. Given the autosomal recessive mode of inheritance of OCA, I think it is an oversight not to describe these additional variants. Are there any computational predictors that suggest biological consequences (regulation, alternative splicing, conservation, CADD score) of these variants, that make it worth mentioning? If so, what are the deep intronic variants? Do the authors consider these variant to be

4. Table 1: What were the specifications for the SpliceAI scores (i.e. masked vs. unmasked, max distance parameters) and what was the predicted effect (acceptor/donor loss/gain) , and delta-bp of the SpliceAI prediction? There are also minor errors with the values in Table 1, none of analytical significance: c.1114G>A SpliceAI: 0.07 (donor loss, -142bp), c.1061T>C Pangolin = 0.05 (Splice Gain); c.1081C>G Pangolin = 0.05 (Splice Gain). These may have been updated in the databases since the time of writing.

5. The allele counts of three of the variants (c.1048G>A, c.1064C>T, c.1109T>C) are notably high. These variants seem to be ancestry-specific as expounded upon briefly in the Discussion in lines 435-436 and Table S4, but I think this information is important enough to describe in the Results in discussion of Table 1.

6. Figure 1C: given that these are RT-PCR samples, these lanes should be labeled with label with genomic variant, not protein variant.

7. There is a typo on line 297. I believe it is supposed to read “haplotype rs1800404-T/rs1800407-T” and not “haplotype rs1800404-T/rs1800404-T”

8. Lines 428-430: the authors recommend minigene assay for novel exon 10 mutations to test for splicing effect. While this may provide nominal research value, testing every exon 10 VUS via minigene is likely not feasible for many clinics or clinical geneticists in routine patient care. Confirming increased exon 10 skipping on a patient level would not alone be enough evidence for a promotion from VUS to Likely Pathogenic or Pathogenic via ACMG guidelines, and it is unclear how it would inform patient care without ASO treatments available.

9. Methods: There is mention of a “local cohort” throughout the manuscript, yet the Materials and Methods section does not detail number of participants or inclusion/exclusion criteria. Assuming this is the same cohort from this group described in reference 12, it would be helpful to (very) briefly describe the cohort and make this overlap clearer in the text.

10. There is a lack of consistency in variant descriptors. The authors switch from HGVS notation and rsID descriptors throughout the manuscript. Notably mentioned in lines 271-272: “c.1065G>A (referred to as rs1800404-T in population genetic studies and here after),” yet c.1065G>A and rs1800404-T are still used interchangeably throughout the manuscript

11. Finally, there are some casual or out of place word choices throughout the manuscript that should be revised. Two examples are in brackets.

a. “Pathogenic variants that cause albinism express themselves as a continuous spectrum from {extreme whiteness} and severe visual impairment to marginally subnormal vision and mild sensitivity of the skin to UV-light.” (lines 53-55)

b. “All animal tissue samples used in this study were acquired from {left-over} post-mortem tissues from other studies involving wild-type control animals” (lines 504-505)

Reviewer #2: The authors present a series of functional experiments in transfected HeLa cells using minitemplates and chimeric constructs of human and mouse OCA2 exon 10 to demonstrate that the ratio of skipping of human OCA2 exon 10 occurs specifically in humans. This processing is modified by the rs1800404 SNP located within that exon. The data is quite convincing. The genetic association studies they conduct between skin and hair phenotypes and the rs1800404 (in exon 10) - rs1800407 (exon 13) haplotypes in the large human collection provided by the UK biobank is sufficient evidence to support the model presented in Figure 8.

Minor comments:

1. In the Introduction the authors state Tyrosinase catalyzes the reaction “…by converting tyrosine into L-Dopa”. It is now understood that Tyrosinase converts tyrosine to dopaquinone (DQ), but that DOPA can assist in that conversion. See Ito and Wakamatsu 2008,

Ito, S., & Wakamatsu, K. (2008). Chemistry of mixed melanogenesis--pivotal roles of dopaquinone. Photochem Photobiol, 84(3), 582-592. doi:10.1111/j.1751-1097.200

2. In the study of the rs1800404-rs1800407 haplotypes in the large collection provided by the UK biobank, is it possible to also do an association study with eye colour? Note that these SNPs were associated with blue eye colour in an earlier study,

Table 3 of Duffy et al, Am J Hum Genet. 2007 Feb;80(2):241-52

A three-single-nucleotide polymorphism haplotype in intron 1 of OCA2 explains most human eye-color variation.

3. In the conducting the analysis of the rs1800404-rs1800407 haplotypes in the UK Biobank cohort, did the authors consider if there was any effect of the major blue/brown eye colour associated SNP rs12913832 on those haplotypes? Is this possible to do rather than be extrapolated in the Discussion. What would be the relative contribution of rs12913832 vs rs1800404/rs1800407 to the penetrance of phenotypes?

4. Is it possible to demonstrate the level/ratio of OCA2 exon10 skipping based on genotypes by RT-PCR directly in RNA extracted from skin tissue or cultured clonal human melanocytic cells.

5. Can the two OCA2 protein isoforms (+/-exon10) be detected and quantitated on protein extracted from tissue or human cultured clonal melanocytic cells based on OCA2 genotype.

**Have all data underlying the figures and results presented in the manuscript been provided?**

Reviewer #1: Yes

Reviewer #2: Yes

PLOS authors have the option to publish the peer review history of their article (what does this mean? ). If published, this will include your full peer review and any attached files.

**Do you want your identity to be public for this peer review?** For information about this choice, including consent withdrawal, please see our Privacy Policy .

Reviewer #1: No

Reviewer #2: No

**Figure resubmission:**
---

## [Editor Report · Decision Letter 1]

28 Aug 2025

Dear Dr Javerzat,

We are pleased to inform you that your manuscript entitled "From paleness to albinism: Contribution of OCA2 exon 10 skipping to hypopigmentation" has been editorially accepted for publication in PLOS Genetics. Congratulations!

Yours sincerely,

Gregory S. Barsh

Academic Editor

PLOS Genetics

Santhosh Girirajan

Section Editor

PLOS Genetics

Aimée Dudley

Editor-in-Chief

PLOS Genetics

Anne Goriely

Editor-in-Chief

PLOS Genetics

Comments from the reviewers (if applicable):

**Data Deposition**

http://datadryad.org/submit?journalID=pgenetics&manu=PGENETICS-D-25-00764R1

**Press Queries**

---

## [Editor Report · Acceptance letter]

PGENETICS-D-25-00764R1

From paleness to albinism: Contribution of OCA2 exon 10 skipping to hypopigmentation

Dear Dr Javerzat,

We are pleased to inform you that your manuscript entitled "From paleness to albinism: Contribution of OCA2 exon 10 skipping to hypopigmentation" has been formally accepted for publication in PLOS Genetics! Your manuscript is now with our production department and you will be notified of the publication date in due course.

With kind regards,

Judit Kozma

PLOS Genetics

On behalf of:
